# SPUR: Scale-Partitioned Uncertainty Rectification for Robust UAV-on-UAV Interception

**Chenqi Yan** [1]  **Zhaoyu Zeng** [1 2]  **Yifeng Yang** [1]  **Jundong Zhou** [1]  **Zhuoyuan Ni** [1]  **Junqi Wu** [1]  **Qinying Gu** [3]
**Xinbing Wang** [1]  **Nanyang Ye** [1 2 3]

## Abstract

Robust aerial target detection for autonomous UAV-on-UAV pursuit is severely hindered by continuous scale drift, long-tailed scale imbalance, and flight-induced visual noise, rendering standard empirical risk minimization strategies poorly aligned with real-world deployment. To address these challenges, we propose a scale-aware robust optimization framework that performs group-wise minimax optimization over scale-partitioned data, ensuring balanced robustness across far-, mid-, and close-range engagement regimes. We further introduce an uncertainty-rectified regression loss to suppress noise-driven errors without discarding informative hard examples, complemented by a control-aligned center accuracy penalty that prioritizes the localization precision required for stable flight control. Extensive experiments demonstrate that our method yields substantially improved robustness under visual degradation, with significantly slower decay in detection mAP and center-point accuracy compared to baselines. Validated through both photorealistic simulations and real-world flight tests, our system achieves **real-time performance of 120 FPS** on an embedded NVIDIA Orin NX platform, confirming its practical efficacy for high-speed interception.

## 1. Introduction

Autonomous UAV-on-UAV interception poses a challenging perception problem, driven by the rapid growth of the low-altitude economy and the increasing prevalence of unauthorized drone flights, where a pursuer must accurately detect and localize a dynamic aerial target using onboard vision under strict latency and computational constraints (Achtelik et al., 2011). Unlike static aerial surveillance, perception errors in this closed-loop interception setting directly propagate to downstream control, potentially destabilizing flight and degrading interception performance (Chaumette & Hutchinson, 2006; Ashraf et al., 2021; Yang et al., 2024). In image-based visual servoing, particularly at long range, even small localization errors can induce large angular deviations, making accurate center-point estimation critical (Chaumette & Hutchinson, 2006).

A central challenge in this problem is systematic scale drift. As the pursuer approaches the target, the apparent target scale varies continuously from a few pixels at long range to a dominant object at close range, a phenomenon induced by interception dynamics rather than dataset bias. Consequently, training data exhibit severe scale imbalance: mid-range targets dominate, while long- and close-range cases form long-tailed but safety-critical regimes. Standard empirical risk minimization (ERM) therefore tends to favor frequent scales and underperform in rare but operationally critical conditions (Sagawa et al., 2020).

This challenge is further compounded by observation noise inherent to aerial platforms. High-speed maneuvers frequently introduce motion blur, while low-light conditions amplify sensor noise (Tang et al., 2023; Du et al., 2019). In such cases, large prediction errors may arise from irreducible aleatoric noise rather than informative hard examples. Consequently, optimization strategies that emphasize high-loss samples can inadvertently amplify noise-dominated outliers, leading to unstable training and degraded robustness (Shrivastava et al., 2016). While existing multi-scale detection methods improve feature representations across scales (Zhao & Zhu, 2023; Jung et al., 2025), they typically do not address the interaction between scale imbalance and noise-induced uncertainty.

In this work, we propose a *scale-partitioned, uncertainty-aware robust optimization framework* for real-time UAV-on-UAV visual tracking. Our approach enforces robustness across target scales via group-wise minimax optimization, while incorporating heteroscedastic uncertainty to attenuate

[1] Shanghai Jiao Tong University, Shanghai, China [2] Shanghai Innovation Institute, Shanghai, China [3] Shanghai AI Laboratory, Shanghai, China. Correspondence to: Nanyang Ye <ynylincoln@sjtu.edu.cn>.

*Proceedings of the 43rd International Conference on Machine Learning*, Seoul, South Korea. PMLR 306, 2026. Copyright 2026 by the author(s).

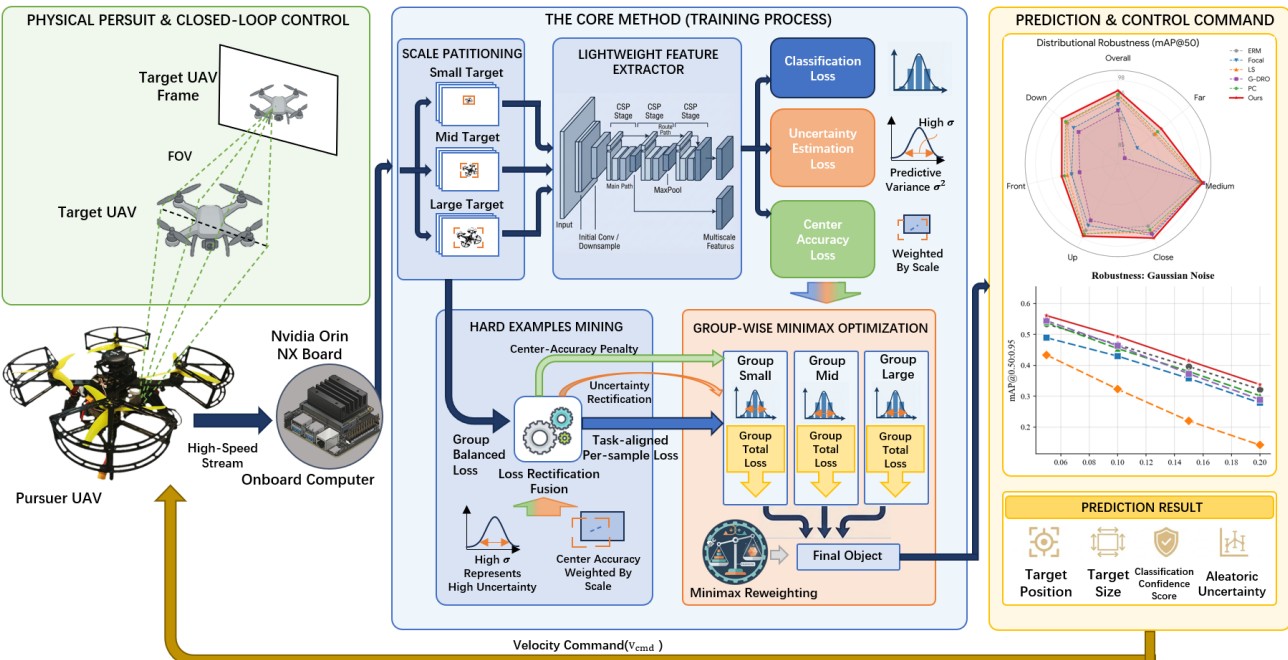

*Figure 1.* **The SPUR Framework.** The training process (center) partitions data into scale-based regimes (small, mid, large). A lightweight feature extractor predicts bounding boxes and aleatoric uncertainty ($\sigma^2$). The Group-wise Minimax Optimization dynamically reweights loss based on worst-case scale groups, while Uncertainty Rectification attenuates gradients from noisy samples (high $\sigma$), and the Control-Aware Penalty enforces tighter localization for distant targets.

gradients from noise-dominated samples. To better align perception with downstream control sensitivity, we further introduce a scale-aware center-point penalty that prioritizes localization stability for small, distant targets. Importantly, our framework operates entirely at the level of the training objective and does not modify the inference architecture, preserving the efficiency of lightweight single-stage detectors for onboard deployment.

We evaluate our method on **UAV-MultiView**, a newly collected dataset featuring diverse UAV platforms, viewpoints, engagement distances, and environmental conditions. Extensive experiments demonstrate improved worst-case performance across scale regimes, enhanced robustness under severe visual corruptions, and stable center-point localization critical for closed-loop control. Deployment on an embedded NVIDIA Orin NX platform confirms real-time performance exceeding **100 FPS**, validating the practicality of the proposed approach.

**Contributions**

- We propose a scale-partitioned uncertainty rectification framework. Extending the detection head with a single uncertainty channel boosts robustness across diverse scales, viewpoints, and corruptions with negligible overhead.
- We construct **UAV-MultiView**, an diverse dataset cap-

turing aerial interception scenarios with rich variations in target platforms and engagement geometries, facilitating future research in this domain.

- We demonstrate the system's practical value through extensive experiments on embedded hardware, achieving superior robustness and high real-time frame rates, validating its suitability for onboard aerial deployment.

## 2. Related Work

### 2.1. Anti-UAV Approaches

Anti-UAV methods generally fall into non-contact or physical-contact categories. Non-contact systems employ signal jamming (Multerer et al., 2017) or GPS spoofing (Kerns et al., 2014) to disrupt control, but suffer from protocol dependence and collateral risks. Physical approaches intercept targets using nets or cooperative capture (Guvenc et al., 2018; Rothe et al., 2019), yet typically rely on human supervision. In contrast, our approach belongs to the physical-contact category but achieves fully autonomous, closed-loop UAV-on-UAV interception without human intervention.

### 2.2. Real-time UAV Detection and Aerial Perception

Due to the high cost and noise sensitivity of radar/RF systems (Hoffmann et al., 2016; Peacock & Johnstone, 2013;

Shao et al., 2022), vision-based detection has become dominant. While two-stage (Ren et al., 2015) and transformer-based models (Zhang, 2023) offer high accuracy, their latency limits real-time usage. Consequently, single-stage YOLO detectors (Redmon et al., 2016; Bochkovskiy et al., 2020; Tian et al., 2026) are preferred for aerial robotics. Although recent works improve detection (Wang et al., 2025; Yao et al., 2025; Wu et al., 2025; Xue et al., 2025), they often prioritize AP over latency, frequently achieving $< 50$ FPS on onboard hardware (Liang et al., 2025). Our work addresses the lack of real-time solutions suitable for agile interception on resource-limited platforms.

## 2.3. Scale-Awareness and Distributional Robustness

Rapid distance changes in interception create severe scale imbalances. While multi-scale aggregation (Wu et al., 2022; Bochkovskiy et al., 2020; Zhao & Zhu, 2023) handles moderate variation, it assumes balanced distributions. Techniques like Distributionally Robust Optimization (DRO) (Sagawa et al., 2020; Duchi & Namkoong, 2021; Jeong et al., 2026) and Online Hard Example Mining (OHEM) (Shrivastava et al., 2016) improve worst-case performance but may overfit noisy samples possessing high aleatoric uncertainty. We address this by synthesizing scale-partitioned minimax optimization with uncertainty rectification. This ensures the model remains robust to under-represented target scales without being distracted by irreducible observation noise, addressing a key limitation of standard GroupDRO in dynamic aerial environments.

## 2.4. Uncertainty Modeling in Visual Regression

Uncertainty is categorized as epistemic (model-based) (Gal & Ghahramani, 2016; Lakshminarayanan et al., 2017; Wang & Ji, 2024; Yelleni et al., 2024) or aleatoric (data-noise) (Kendall & Gal, 2017; He et al., 2019; Choi et al., 2019). Since interception imagery is dominated by motion blur and sensor noise rather than data scarcity, we focus on aleatoric uncertainty. Instead of using it merely for calibration, we leverage it as a structural component within a minimax objective, preventing robust optimization from amplifying noisy samples.

## 2.5. Perception–Control Alignment in Vision Systems

Standard vision metrics like IoU often misalign with downstream control requirements (Philion et al., 2020; Hu et al., 2023; Codevilla et al., 2018). In Image-Based Visual Servoing (IBVS), center-point stability is critical, as small pixel-level errors at long range induce significant instability (Chaumette & Hutchinson, 2006). We address this by incorporating a scale-aware center-point penalty, strictly aligning visual learning objectives with UAV control dynamics.

## 3. Methodology

We consider UAV-UAV visual tracking in an interception setting, where a pursuer UAV must continuously localize and approach an adversarial aerial target using onboard vision. Unlike static object detection, this problem exhibits a strong coupling between perception, data distribution, and downstream control. As the UAV closes the distance, the apparent scale of the same target evolves continuously, while the visual observations are simultaneously corrupted by flight-induced noise such as motion blur, vibration, and atmospheric distortion. These characteristics expose a fundamental mismatch between standard detector training objectives and the operational requirements of aerial tracking.

## 3.1. Scale-Partitioned Robust Optimization

From a physical perspective, the interception process induces a *structured distribution drift* over target scales: long-range targets dominate early pursuit and occupy only a few pixels, whereas close-range targets appear briefly but are critical for terminal control. In practice, training datasets reflect this imbalance, with mid-scale samples being abundant and extreme scales forming long-tailed, underrepresented states. Under empirical risk minimization, detectors therefore overfit to dominant scales and fail to generalize reliably to rare but safety-critical regimes.

To address this, we formulate training as a scale-partitioned minimax optimization problem. Unlike standard Group-DRO which typically partitions data by categorical attributes within each batch, we explicitly partition the training set into $K$ disjoint subsets $\{S_k\}_{k=1}^K$ based on the target's scale in the image frame (e.g., small, mid, and large targets) using the bounding box area $A_i$ as a proxy. We then seek parameters robust to the worst-case distribution within each scale regime via the following objective:

$$L_{\text{Hybrid}}(\theta) = \frac{1}{K} \sum_{k=1}^{K} \left( \max_{p_k \in \mathcal{P}_k} \sum_{(x_i, y_i) \in S_k} p_{k,i} \, \mathcal{L}_i^{\text{Task}}(\theta) \right),$$

(1)

where $\mathcal{P}_k = \{p_k \mid D_{\text{KL}}(p_k \| p_{k,0}) \leq \rho\}$ denotes a KL-divergence uncertainty set around the empirical distribution $p_{k,0}$. This formulation ensures that each scale regime contributes equally to the optimization, preventing frequent mid-range samples from overwhelming rare but operationally critical cases.

## 3.2. Aleatoric Uncertainty Rectification under Minimax Reweighting

While scale-partitioned minimax optimization improves robustness across distance regimes, it introduces a critical challenge in aerial vision: sensitivity to noise. UAV-mounted cameras frequently suffer from motion blur, vibration, and

atmospheric effects, particularly during aggressive maneuvers. In such settings, large losses often arise not from informative hard examples but from irreducible aleatoric uncertainty. Under the minimax objective in Eq. (1), these samples can be aggressively upweighted, potentially amplifying noise and destabilizing training.

To prevent this failure mode, we explicitly incorporate *heteroscedastic uncertainty modeling* into the loss definition (Kendall & Gal, 2017). Unlike (He et al., 2019) which utilizes variance solely for bounding box refinement, we leverage uncertainty as a dynamic reliability indicator within our Minimax objective. Specifically, we predict a learnable scalar $s_i = \log(\sigma_i^2)$ for each anchor to model the isotropic aleatoric uncertainty. Consequently, the regression component is reformulated as an uncertainty-attenuated error energy rather than a strict log-likelihood:

$$\mathcal{L}_{\text{Uncertainty}}(i) = \exp(-s_i)\mathcal{L}_{\text{reg}}(\hat{y}_i, y_i) + \frac{1}{2}s_i. \quad (2)$$

This formulation naturally attenuates the influence of inherently noisy samples by exponentially down-weighting their regression errors, while the regularization term $\frac{1}{2}s_i$ prevents the model from trivially predicting infinite uncertainty. Crucially, when combined with minimax reweighting, this mechanism ensures that gradients are dominated by samples that are structurally difficult yet reliable, rather than those corrupted by sensor noise.

### 3.3. Control-Aware Localization Objective

Scale robustness and noise robustness alone are insufficient for stable aerial interception, as visual localization errors propagate directly into the UAV control loop. The impact of these errors depends strongly on target scale: for distant targets, even sub-pixel deviations in the predicted bounding box center can induce large angular errors, leading to oscillatory or unstable behavior. Standard IoU-based losses fail to capture this asymmetry.

To align perception with control sensitivity, we introduce a scale-aware center-point penalty:

$$\mathcal{L}_{\text{Control}}(i) = w(A_i) \|\hat{c}_i - c_i\|_2^2, \quad (3)$$

where $(\hat{c}_i, c_i)$ denote the predicted and ground-truth box centers, and

$$w(A_i) = \frac{1}{A_i + \epsilon}, \quad (4)$$

assigns larger penalties to small targets. This term explicitly prioritizes center-point stability in long-range tracking, where control sensitivity is highest.

### 3.4. Decoupled Minimax Optimization

A critical conflict arises when integrating the uncertainty mechanism (Eq. (2)) into the scale-partitioned optimization

(Eq. (1)). If the adversarial weights $p_{k,i}$ are computed using the uncertainty-attenuated loss, the adversary fails. Specifically, for a structurally hard example, the model can simply predict high uncertainty to reduce the loss magnitude, thereby "hiding" the sample from the adversary. Consequently, the Minimax mechanism fails to mine informative hard examples.

To resolve this dilemma, we propose a **Decoupled Minimax Strategy**. We separate the loss function into two distinct components: a *Proxy Loss* for the adversary (to measure difficulty), and an *Optimization Loss* for the predictor (to update parameters).

**1. The Proxy Loss ($\mathcal{L}^{\textbf{Proxy}}$):** This component represents the raw task performance without uncertainty attenuation. It serves as the uncorrupted signal for the adversary to identify hard samples.

$$\mathcal{L}_i^{\text{Proxy}} = \mathcal{L}_{\text{cls}}(i) + \mathcal{L}_{\text{reg}}(\hat{y}_i, y_i) + \lambda_2 \mathcal{L}_{\text{Control}}(i). \quad (5)$$

Note that $\mathcal{L}_{\text{reg}}$ here denotes the standard regression loss (e.g., CIoU) without the $\exp(-s_i)$ scaling.

**2. The Optimization Loss ($\mathcal{L}^{\textbf{Optim}}$):** This component includes the uncertainty rectification and regularization terms, used for the actual parameter update via backpropagation.

$$\mathcal{L}_i^{\text{Optim}} = \mathcal{L}_{\text{cls}}(i) + \lambda_1 \mathcal{L}_{\text{Uncertainty}}(i) + \lambda_2 \mathcal{L}_{\text{Control}}(i). \quad (6)$$

We reformulate the optimization by computing the adversarial weights $p_{k,i}^*$ based on the Proxy Loss, while applying them to the Optimization Loss. The closed-form solution for the worst-case distribution within group $k$ is given by:

$$p_{k,i}^* = \frac{\exp(\mathcal{L}_i^{\text{Proxy}}/\eta)}{\sum_{j \in S_k} \exp(\mathcal{L}_j^{\text{Proxy}}/\eta)}. \quad (7)$$

Crucially, $p_{k,i}^*$ is treated as a constant (detached from the computation graph) with respect to the uncertainty parameters $s_i$. The final gradient update is:

$$\nabla_\theta \mathcal{L}_{\text{Final}} = \sum_{k=1}^{K} \sum_{(x_i,y_i) \in S_k} \text{stop\_grad}(p_{k,i}^*) \cdot \nabla_\theta \mathcal{L}_i^{\text{Optim}}. \quad (8)$$

This decoupling strategy prevents the model from evading hard-example mining via uncertainty inflation, as adversarial weights are strictly determined by the raw proxy error. By isolating the uncertainty rectification to the gradient update step, our framework enforces a "strict selection, flexible update" paradigm: it ensures that the optimizer aggressively targets structurally difficult samples while retaining the capacity to dampen gradients from irreducible sensor noise, effectively balancing scale robustness with noise tolerance.

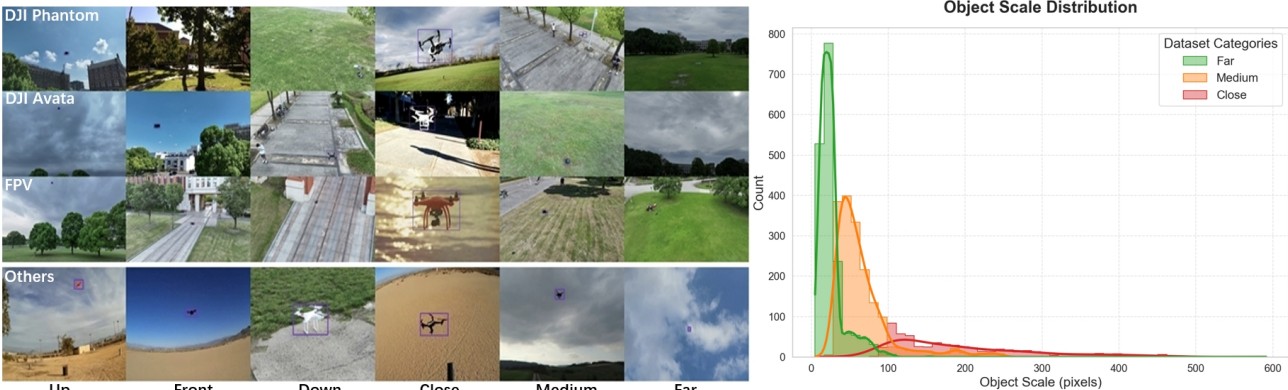

*Figure 2.* **Overview of the UAV-MultiView Dataset. (Left)** Representative samples showcasing the diversity of target platforms (e.g., DJI Phantom, Avata, FPV), environmental contexts, and relative viewing geometries (Up, Front, Down). **(Right)** The object scale distribution (in pixels) of test dataset across the three engagement regimes (Far, Medium, Close). The histogram highlights the severe scale imbalance and long-tailed distribution inherent to aerial interception, which motivates our scale-partitioned optimization approach.

# 4. Experiments

## 4.1. Experimental Setup

**Dataset.** We evaluate our framework on **UAV-MultiView**, a curated benchmark designed to assess robustness against systematic scale and viewpoint variations. Existing public datasets often suffer from specific distributional biases, predominantly focusing on close-range or top-down surveillance views. They frequently lack the diverse camera-target geometries and long-range samples characterizing real-world UAV interception scenarios. To address this gap, we integrate samples from multiple public sources (Pawełczyk & Wojtyra, 2020; Zhao et al., 2022; Chen et al., 2017) and augment them with self-collected aerial imagery captured by a DJI Mavic 3 Pro, totaling 20,250 annotated images.

As illustrated in Fig. 2, UAV-MultiView comprises a total of **20,250** high-quality labeled images explicitly structured along two geometric axes. Regarding *viewpoint*, the dataset contains 6,051 downward, 5,385 frontal, and 8,814 upward samples. In terms of *target distance*, it includes 8,531 far, 8,908 medium, and 2,811 close-range images. For evaluation, the data is randomly partitioned into training, validation, and test sets with a 7:1:2 ratio. We further construct stratified test subsets based on these geometric attributes to enable fine-grained robustness analysis.

**Metrics.** We report standard Mean Average Precision (mAP) for detection performance. Additionally, we introduce Center MAE, defined as the mean absolute pixel error between predicted and ground-truth box centers. This metric serves as a proxy for downstream control accuracy, as center deviations directly impact angular tracking errors in UAV interception loops.

**Implementation Details.** All models utilize a lightweight CSPDarknet backbone (aligned with YOLO-tiny) and are deployed on an NVIDIA Orin NX (FP16). Training employs the Adam optimizer with an initial learning rate of $1 \times 10^{-3}$ and cosine annealing decay. Models are trained for 300 epochs, with convergence typically observed between 100–150 epochs. Consistent Mixup and Mosaic augmentations are applied across all experiments.

For comparative analysis, we implement the following robust baselines with optimized hyperparameters: (1) **Focal Loss** (Lin et al., 2017) with $\alpha = 0.25$ and $\gamma = 2.0$; (2) **Label Smoothing (LS)** (Szegedy et al., 2016) with a smoothing factor $\epsilon = 0.01$; (3) **CVaR-style DRO variant** (Duchi & Namkoong, 2021) with a weight ratio of 0.2; (4) **Penalized Confidence (PC)** (Pereyra et al., 2017) with a regularization weight of 0.1. For our method, the spectral clustering hyperparameter is set to $K = 3$.

## 4.2. Distributional Robustness Across Scale and Viewpoint Groups

To evaluate robustness under structured distribution shifts induced by target scale and viewing geometry, we report performance on stratified test subsets defined by target distance (Far, Medium, Close) and relative viewpoint (Up, Front, Down). These subsets correspond to subpopulations within the same dataset and reflect operational regimes with highly imbalanced sample frequencies.

As reported in Table 1, standard ERM exhibits a distinct performance degradation when facing distribution shifts, particularly on subsets with extreme geometric variations. For instance, while ERM attains a high 97.96% mAP$_{50}$ on *Medium* scale targets, its accuracy drops sharply to 90.98% on the *Far* subset, where reduced pixel resolution and feature ambiguity pose significant challenges.

In contrast, our method demonstrates superior distributional robustness across both scale and viewpoint dimensions. We

*Table 1.* Distributional robustness across scale and viewpoint groups. Performance is reported on stratified test subsets within the same dataset. We compare our method against Focal Loss, Label Smoothing, CVaR-DRO and Penalize confidence. Our approach achieves the best balance across all scenarios. Best results are **bolded**, and second-best results are underlined.

| Test Subset | Metric | ERM | Focal | LS | CVaR-DRO | PC | Ours |
|---|---|---|---|---|---|---|---|
| Test Set (Overall) | $\text{mAP}_{50} \uparrow$ | 94.64 | 93.48 | 95.25 | 92.25 | 95.35 | **96.10** |
| | $\text{mAP}_{50:95} \uparrow$ | 56.72 | 51.91 | 58.19 | 48.56 | 58.66 | **60.49** |
| *Scale Groups* | | | | | | | |
| Test Far (Small) | $\text{mAP}_{50} \uparrow$ | 90.98 | 86.76 | 91.29 | 83.66 | 91.84 | **92.75** |
| | $\text{mAP}_{50:95} \uparrow$ | 47.02 | 40.51 | 47.80 | 35.55 | **48.91** | 48.62 |
| Test Medium | $\text{mAP}_{50} \uparrow$ | 97.96 | 98.70 | 98.50 | **98.97** | 98.67 | 98.80 |
| | $\text{mAP}_{50:95} \uparrow$ | 66.39 | 59.57 | 66.77 | 58.50 | 67.13 | **69.88** |
| Test Close (Large) | $\text{mAP}_{50} \uparrow$ | 95.45 | 97.43 | 96.83 | 97.06 | 96.25 | **98.01** |
| | $\text{mAP}_{50:95} \uparrow$ | 66.02 | 62.43 | 68.27 | 62.16 | 64.19 | **73.56** |
| *Viewpoint Groups* | | | | | | | |
| Test Up | $\text{mAP}_{50} \uparrow$ | 96.36 | 95.33 | 97.03 | 94.21 | 97.13 | **97.58** |
| | $\text{mAP}_{50:95} \uparrow$ | 56.99 | 52.96 | 58.18 | 49.66 | 58.75 | **59.60** |
| Test Front | $\text{mAP}_{50} \uparrow$ | 91.23 | 91.22 | 92.06 | 89.62 | 92.66 | **93.18** |
| | $\text{mAP}_{50:95} \uparrow$ | 54.31 | 49.30 | 56.09 | 45.42 | 56.23 | **59.22** |
| Test Down | $\text{mAP}_{50} \uparrow$ | 94.50 | 93.02 | 94.80 | 91.76 | 94.98 | **95.93** |
| | $\text{mAP}_{50:95} \uparrow$ | 58.76 | 53.16 | 59.84 | 49.95 | 60.67 | **62.88** |

achieve the best overall performance with a state-of-the-art 96.10% $\text{mAP}_{50}$ and 60.49% $\text{mAP}_{50:95}$, outperforming strong baselines including Penalized Confidence and CVaR-DRO. Specifically on the critical *Far* group, our approach recovers a substantial margin, boosting $\text{mAP}_{50}$ to 92.75%. Notably, these gains on hard subgroups (e.g., *Front* and *Down* viewpoints) are achieved without compromising performance on dominant regimes, as evidenced by our top-tier results on *Close* and *Medium* targets.

These results validate that explicitly modeling scale-induced shifts effectively mitigates the bias of ERM toward average cases. By ensuring consistent high precision across all stratified subsets, our framework is better equipped to handle the drastic scale and perspective changes inherent to dynamic UAV interception.

### 4.3. Robustness to Environmental Corruptions

Robust UAV interception requires reliable perception under severe sensor degradation caused by motion, vibration, and atmospheric disturbances. To evaluate robustness under such conditions, we conduct controlled stress tests by applying increasing levels of Gaussian noise and motion blur to the test set, following common corruption-based robustness protocols. All corruptions are applied only at test time; models are trained on clean data. Results are summarized in Table 2.

As detailed in Table 2, standard ERM suffers distinct performance degradation as corruption severity increases. Under heavy Gaussian noise ($\sigma = 0.20$), ERM's detection capability drops to 67.28% $\text{mAP}_{50}$ with a coarse 5.37-pixel localization error. In contrast, our method demonstrates

superior resilience, recovering accuracy to 68.57% $\text{mAP}_{50}$ while significantly tightening the localization error to 4.79 pixels.

A similar trend is observed under motion blur. While Label Smoothing proves competitive in detection metrics at high intensities ($k \geq 7$), our approach dominates in localization stability, achieving the lowest Center MAE across all kernel sizes. Notably, at $k = 9$, our method maintains a tracking error of just 4.35 pixels, despite the severe visual smearing.

These results validate that our method effectively attenuates gradient noise from corrupted features. By prioritizing high-confidence regression, our framework ensures precise target localization even when general recognition features are degraded—a critical attribute for reliable UAV interception in turbulent environments.

### 4.4. Onboard Efficiency and Real-World Feasibility

Since our modifications are strictly limited to the training objective, the inference architecture remains identical to the efficient lightweight baseline, incurring negligible inference cost. We deploy the model on an embedded NVIDIA Orin NX Developer Kit (base) using TensorRT (INT8) optimization. As shown in Table 3, the system achieves **87–121 FPS** on embedded hardware. This high throughput minimizes perception latency, ensuring stable feedback control even under aggressive maneuvers and high-speed engagements.

To validate the practical feasibility of this high-speed pipeline, we conducted a staged evaluation. We first verified the stability of the perception-control interface in high-fidelity simulations and subsequently deployed the system

*Table 2.* Comparison of robustness under Gaussian noise and motion blur. We report **mAP$_{50}$** (%, higher is better) and **Center MAE** (pixels, lower is better). **Bold** and underlined indicate the best and second-best results.

| Noise Type | Metric | ERM | Focal | CVaR-DRO | LS | PC | Ours |
|---|---|---|---|---|---|---|---|
| *Gaussian Noise* | | | | | | | |
| $\sigma = 0.05$ | mAP$_{50}$ ↑ | 92.60 | 91.06 | 88.44 | 92.33 | 92.60 | **93.42** |
| | Center MAE ↓ | 4.06 | 4.40 | 4.60 | 3.67 | 3.81 | **3.31** |
| $\sigma = 0.10$ | mAP$_{50}$ ↑ | 86.36 | 86.80 | 76.96 | 85.67 | 85.62 | **87.69** |
| | Center MAE ↓ | 4.34 | 4.83 | 5.37 | 4.27 | 4.33 | **3.77** |
| $\sigma = 0.15$ | mAP$_{50}$ ↑ | 77.35 | 77.79 | 61.90 | 76.61 | 75.94 | **79.53** |
| | Center MAE ↓ | 4.91 | 5.58 | 5.94 | 4.84 | 4.88 | **4.36** |
| $\sigma = 0.20$ | mAP$_{50}$ ↑ | 67.28 | 66.69 | 46.42 | 66.33 | 64.21 | **68.57** |
| | Center MAE ↓ | 5.37 | 6.28 | 6.27 | 5.41 | 5.42 | **4.79** |
| *Motion Blur* | | | | | | | |
| $k = 3$ | mAP$_{50}$ ↑ | 93.37 | 92.53 | 90.34 | 94.05 | 93.87 | **94.94** |
| | Center MAE ↓ | 3.80 | 4.26 | 4.46 | 3.61 | 3.82 | **3.33** |
| $k = 5$ | mAP$_{50}$ ↑ | 90.26 | 88.46 | 86.41 | 91.59 | 89.95 | **91.72** |
| | Center MAE ↓ | 3.96 | 4.56 | 4.88 | 3.90 | 4.09 | **3.60** |
| $k = 7$ | mAP$_{50}$ ↑ | 84.42 | 81.87 | 80.27 | **85.93** | 82.40 | 84.29 |
| | Center MAE ↓ | 4.30 | 5.08 | 5.31 | 4.25 | 4.64 | **3.96** |
| $k = 9$ | mAP$_{50}$ ↑ | 76.03 | 73.39 | 71.03 | **77.09** | 71.31 | 75.37 |
| | Center MAE ↓ | 4.59 | 5.58 | 5.88 | 4.77 | 5.27 | **4.35** |

*Table 3.* Inference latency on NVIDIA Orin NX. Our method uses the same architecture, thus sharing these acceleration benefits.

| Precision | Power | Latency | FPS |
|---|---|---|---|
| PyTorch | 15W | 47.55 ms | 21.18 |
| TensorRT (FP16) | 15W | 17.36 ms | 59.59 |
| TensorRT (INT8) | 15W | 15.45 ms | 67.60 |
| TensorRT (INT8) | 25W | 11.57 ms | 87.77 |
| TensorRT (INT8) | MAXN | **8.34 ms** | **121.07** |

on a physical UAV for real-world flight tests. Qualitative results, visualized in Figure 3, demonstrate that the detector maintains high recognition accuracy and localization stability under real-world dynamics. This confirms that our framework successfully bridges the gap between robust optimization and real-time onboard control constraints.

## 5. Ablation Studies

We perform comprehensive ablation studies to isolate the contributions of our framework's key components and verify their interaction effects. We evaluate five variants: (i) standard ERM, (ii) Group-wise Minimax (without uncertainty), (iii) a Coupled Group-wise Minimax+Uncertainty strategy, (iv) our proposed Decoupled strategy, and (v) the **Full** method with control-aware regularization.

### 5.1. Scale Robustness and The Necessity of Decoupling

Table 4 highlights the impact of optimization strategies on scale imbalance. Standard ERM shows a clear bias towards dominant *Medium* targets (97.96%), degrading significantly

on *Far* targets (90.98%). Introducing Group-wise Minimax (Row 2) effectively mitigates this bias, improving worst-case performance to 91.95% by enforcing uniform risk across scales.

**The "Evasion" Phenomenon in Coupled Optimization.** A critical finding is the failure of the Coupled variant (Row 3). Naively integrating uncertainty into the Minimax objective leads to a performance collapse, particularly on *Far* targets (86.76%), falling well below the ERM baseline. This empirical evidence strongly validates our theoretical analysis in Section 3: when coupled, the uncertainty term acts as a "shortcut" for the optimizer. Instead of learning robust features for hard samples, the model simply inflates the predicted variance $\sigma$ to reduce the weighted loss. This "evasion" strategy effectively hides hard samples from the adversary, rendering the Minimax mechanism futile.

**Restoration via Decoupling.** Our Decoupled strategy (Row 4) successfully resolves this conflict. By utilizing the raw proxy loss for adversarial weighting while keeping uncertainty for gradient rectification, we force the model to confront hard samples directly. This restores the worst-group performance (92.18% on *Far*), proving that uncertainty must be treated as a gradient rectifier rather than a weighting factor.

### 5.2. Noise Robustness and Control Alignment

We next evaluate robustness under environmental corruptions (Table 5). While Group-wise Minimax improves scale balance, it introduces a side effect: sensitivity to noise. As shown in Row 2, it underperforms ERM under Gaussian

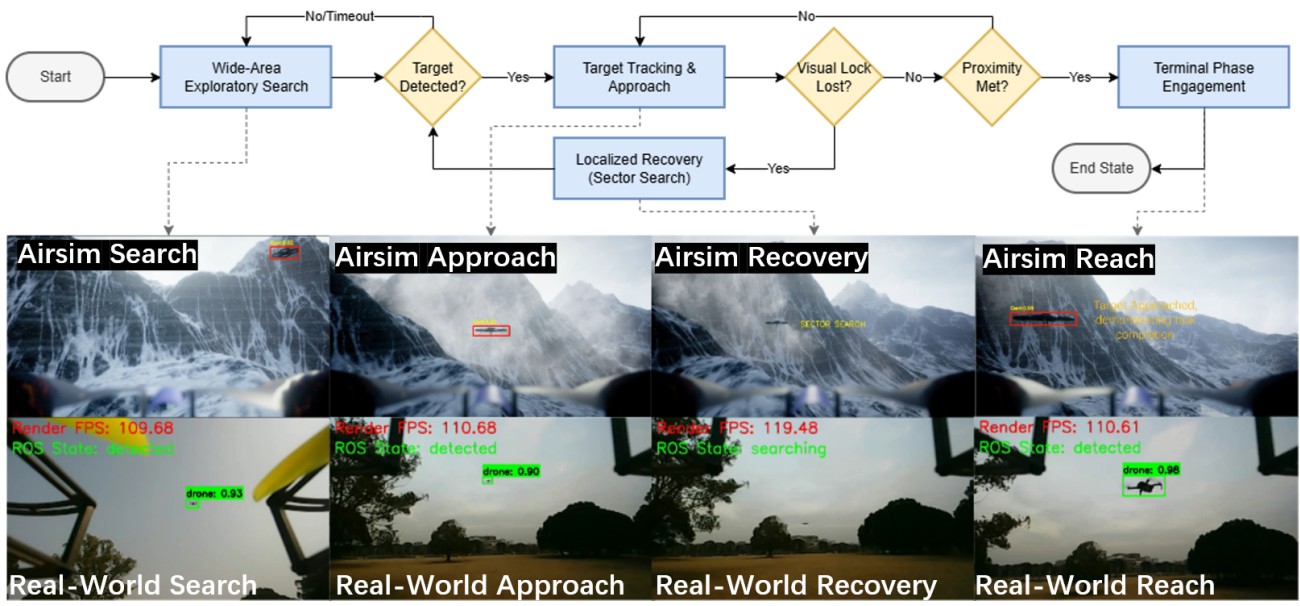

*Figure 3.* **Autonomous Interception Pipeline and Flight Tests.** Top: The state machine logic switching between search, approach, and terminal engagement. Bottom: Qualitative results comparing AirSim simulation (top row) with real-world flight tests (bottom row) on an NVIDIA Orin NX, demonstrating stable tracking during high-speed maneuvers.

*Table 4.* Ablation study on scale robustness. We report mAP@0.50 (%), where higher is better.

| Method | Far | Med. | Close | All |
|---|---|---|---|---|
| ERM | 90.98 | 97.96 | 95.45 | 94.64 |
| Group-wise Minimax | 91.95 | 98.49 | 96.97 | 95.51 |
| Minimax+Uncertainty (Coupled) | 86.76 | 98.70 | 97.43 | 93.99 |
| Minimax+Uncertainty (Decoupled) | 92.18 | 98.69 | 97.56 | 95.80 |
| Ours (Full) | **92.75** | **98.80** | **98.01** | **96.10** |

*Table 5.* Ablation study on robustness under environmental corruptions. We report mAP@0.50 (%, higher is better) and Center MAE (pixels, lower is better) under representative noise levels.

| Method | $mAP_{50}$ | CMAE |
|---|---|---|
| *Gaussian Noise, $\sigma = 0.05$* | | |
| ERM | 92.60 | 4.0554 |
| Group-wise Minimax | 90.84 | 3.5395 |
| Minimax+Uncertainty (Coupled) | 91.06 | 4.4037 |
| Minimax+Uncertainty (Decoupled) | 93.34 | 3.3168 |
| Ours (Full) | **93.42** | **3.3061** |
| *Motion Blur, $k = 5$* | | |
| ERM | 90.26 | 3.9567 |
| Group-wise Minimax | 89.57 | 3.9072 |
| Minimax+Uncertainty (Coupled) | 88.46 | 4.5565 |
| Minimax+Uncertainty (Decoupled) | 91.43 | 3.6109 |
| Ours (Full) | **91.72** | **3.6001** |

noise (90.84% vs 92.60%), indicating that the adversary aggressively upweights aleatoric outliers. Incorporating uncertainty rectification via our Decoupled strategy corrects this behavior, distinguishing between structural difficulty and sensor noise, which yields a substantial gain of +2.5% mAP over the pure Minimax baseline.

**Geometric Alignment via Control-Aware Loss.** Finally, the Full method (Row 5) integrates control-aware regularization. While the overall mAP improvement appears modest (+0.3%), this metric belies the component's true value. Standard detection metrics like IoU are area-dominated and relatively insensitive to minor center shifts, yet these shifts translate to significant angular errors in flight control. The proposed loss targets this geometric disconnect. By achieving the lowest Center MAE (3.3061 px) and further boosting *Far* regime accuracy (92.75%), the full method ensures that perception outputs are not just visually plausible but *control-ready*, reducing potential oscillations during the critical initial interception phase.

## 6. Conclusion

We propose a scale-partitioned minimax framework that decouples adversarial reweighting from uncertainty rectification to effectively handle scale imbalance and sensor noise in UAV interception. By rigorously preventing the optimization collapse observed in coupled objectives and integrating a control-aware objective, our approach achieves superior robustness under environmental corruptions compared to standard baselines. Real-time benchmarks on embedded hardware, supported by our open-source UAV-MultiView dataset, validate its feasibility for high-dynamic, autonomous aerial engagements.

## Acknowledgements

This work is supported by New Generation Artificial Intelligence-National Science and Technology Major Project (No.2025ZD0122901). This work is also supported by National Natural Science Foundation of China (No.62572313, No.62106139).

## Impact Statement

This paper studies robust visual perception under challenging conditions, motivated by safety-critical applications such as protecting civilian airspace and critical infrastructure from unauthorized drones. But we acknowledge that such perception technologies are inherently dual-use. To mitigate these risks, any public release will be limited to research-oriented with explicit restrictions against non-research usage.

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

# A. Theoretical Analysis of Gradient Dynamics

In this section, we provide a formal analysis of the gradient dynamics underlying the proposed decoupled optimization framework. Uncertainty modeling is essential for handling aleatoric noise and intrinsic ambiguity in dense prediction tasks, as it attenuates gradient contributions arising from irreducible errors and improves robustness. However, when uncertainty is directly coupled with the minimax objective, it inadvertently becomes a mechanism for suppressing gradients from genuinely hard but informative samples, leading to *gradient dampening* and compressing the intended exponential hard-mining signal into a polynomial one. In contrast, by decoupling uncertainty modeling from adversarial reweighting, the proposed strategy preserves the robustness benefits of uncertainty estimation while maintaining the exponential sensitivity required for effective hard-example mining.

## A.1. Preliminaries

Let $\mathcal{L}_{\text{raw}}(\theta; x_i)$ denote the raw task loss (e.g., the regression component) for a sample $x_i$ parameterized by $\theta$. Following the Gaussian negative log-likelihood formulation in the main text, let $s_i := \log(\sigma_i^2)$ be the predicted log-variance for sample $x_i$. The aleatoric uncertainty-aware loss is defined as:

$$\mathcal{L}_{\text{unc}}(\theta, s_i; x_i) = \frac{1}{2} \exp(-s_i)\mathcal{L}_{\text{raw}}(\theta; x_i) + \frac{1}{2} s_i. \tag{9}$$

In the minimax framework, the adversarial weight $p_i^*$ for a sample $x_i$ is governed by the Boltzmann distribution based on a signal loss $\ell_i$:

$$p_i^*(\ell) \propto \exp\left(\frac{\ell_i}{\eta}\right), \tag{10}$$

where $\eta$ is the temperature hyperparameter. We analyze the behavior of the gradient magnitude $\|\nabla_\theta J\|$ as the raw error $\mathcal{L}_{\text{raw}}$ becomes large (i.e., for "hard" examples).

## A.2. Gradient Analysis of Coupled Optimization

**Proposition A.1.** *(Gradient Dampening in Coupled Setting). If the adversarial weight is computed using the uncertainty-rectified loss (i.e., $\ell_i = \mathcal{L}_{unc}$), the adversarial weighting mechanism degenerates from exponential to polynomial scaling with respect to the raw error.*

*Proof.* Consider the optimal uncertainty response $s_i^*$ for a fixed raw loss $\mathcal{L}_{\text{raw}}(\theta; x_i)$. Minimizing Eq. 9 w.r.t. $s_i$ yields the stationarity condition:

$$\frac{\partial \mathcal{L}_{\text{unc}}}{\partial s_i} = -\frac{1}{2} \exp(-s_i)\mathcal{L}_{\text{raw}} + \frac{1}{2} = 0 \implies s_i^* = \log(\mathcal{L}_{\text{raw}}). \tag{11}$$

Substituting $s_i^*$ back into Eq. 9, the optimal rectified loss becomes:

$$\mathcal{L}_{\text{unc}}^* = \frac{1}{2} \exp(-\log \mathcal{L}_{\text{raw}})\mathcal{L}_{\text{raw}} + \frac{1}{2} \log(\mathcal{L}_{\text{raw}}) = \frac{1}{2}\left(1 + \log \mathcal{L}_{\text{raw}}\right). \tag{12}$$

The gradient of the total objective $J$ with respect to the feature parameters $\theta$ is:

$$\nabla_\theta J = p_i^*(\mathcal{L}_{\text{unc}}^*) \cdot \nabla_\theta \mathcal{L}_{\text{unc}}\big|_{s_i = s_i^*}. \tag{13}$$

The local gradient term at the optimal uncertainty state is:

$$\nabla_\theta \mathcal{L}_{\text{unc}}\big|_{s_i = s_i^*} = \frac{1}{2} \exp(-s_i^*)\nabla_\theta \mathcal{L}_{\text{raw}} = \frac{1}{2\mathcal{L}_{\text{raw}}}\nabla_\theta \mathcal{L}_{\text{raw}}. \tag{14}$$

Critically, due to the logarithmic scaling of $\mathcal{L}_{\text{unc}}^*$ (Eq. 12), the adversarial weight $p_i^*$ scales polynomially rather than exponentially:

$$p_i^* \propto \exp\left(\frac{1 + \log \mathcal{L}_{\text{raw}}}{2\eta}\right) = C \cdot (\mathcal{L}_{\text{raw}})^{\frac{1}{2\eta}}. \tag{15}$$

Combining these terms, the effective gradient scales as:

$$\|\nabla_\theta J\| \propto (\mathcal{L}_{\text{raw}})^{\frac{1}{2\eta}} \cdot \frac{1}{\mathcal{L}_{\text{raw}}}\|\nabla_\theta \mathcal{L}_{\text{raw}}\| = (\mathcal{L}_{\text{raw}})^{\frac{1}{2\eta} - 1}\|\nabla_\theta \mathcal{L}_{\text{raw}}\|. \tag{16}$$

For standard settings where $\eta \geq 1$, the exponent $(\frac{1}{2\eta} - 1)$ is negative. Unless $\|\nabla_\theta \mathcal{L}_{\text{raw}}\|$ grows super-linearly, the gradient contribution is significantly dampened. This analysis highlights a fundamental conflict: while uncertainty is intended to attenuate gradients arising from irreducible noise, coupling it directly into the minimax objective allows it to also suppress gradients from genuinely hard-but-informative samples, thereby undermining the purpose of adversarial hard mining.

### A.3. Gradient Analysis of Decoupled Optimization

The decoupled strategy resolves this conflict by assigning uncertainty modeling and hard-sample reweighting to separate roles: uncertainty governs the local robustness of the loss, while the minimax objective operates on a raw error proxy that reflects epistemic difficulty.

**Proposition A.2.** *(Sensitivity Preservation in Decoupled Setting). If the adversarial weight is computed using the raw proxy loss (i.e., $\ell_i = \mathcal{L}_{raw}$), the effective gradient magnitude maintains exponential sensitivity to hard examples.*

*Proof.* In the decoupled strategy, the adversarial weight depends directly on $\mathcal{L}_{\text{raw}}$:

$$p_i^* \propto \exp\left(\frac{\mathcal{L}_{\text{raw}}}{\eta}\right). \tag{17}$$

The parameter update is still performed on $\mathcal{L}_{\text{unc}}$ to retain aleatoric noise robustness. The total gradient is:

$$\nabla_\theta J_{\text{decoupled}} = p_i^*(\mathcal{L}_{\text{raw}}) \cdot \nabla_\theta \mathcal{L}_{\text{unc}}. \tag{18}$$

Assuming optimal uncertainty estimation ($s_i^* = \log \mathcal{L}_{\text{raw}}$), the gradient scales as:

$$\|\nabla_\theta J_{\text{decoupled}}\| \propto \exp\left(\frac{\mathcal{L}_{\text{raw}}}{\eta}\right) \cdot \frac{1}{2\mathcal{L}_{\text{raw}}}\|\nabla_\theta \mathcal{L}_{\text{raw}}\|. \tag{19}$$

Comparing Eq. 16 and Eq. 19, the decoupled term contains the exponential factor $\exp(\mathcal{L}_{\text{raw}}/\eta)$, which dominates the polynomial denominator. As $\mathcal{L}_{\text{raw}}$ increases, the gradient magnitude grows exponentially, ensuring that the optimizer prioritizes reducing the epistemic error $\mathcal{L}_{\text{raw}}$ despite the uncertainty attenuation. $\square$

### A.4. Remark on Loss Functions

**Bounded vs. Unbounded Losses:** While specific loss functions employed in implementation (e.g., CIoU) may be bounded, the analysis above assumes a generic unbounded loss landscape to characterize the asymptotic behavior of the optimization dynamics. The derived scaling laws demonstrate that for samples with large errors (relative to $\eta$), the coupled approach suffers from saturation, whereas the decoupled approach maintains active re-weighting.

**Composite Loss:** The total optimization loss includes components not attenuated by uncertainty (e.g., classification loss). The analysis in Propositions A.1 and A.2 focuses specifically on the uncertainty-sensitive branches (e.g., bounding box regression). The decoupled strategy ensures that these specific branches cannot bypass the minimax objective via uncertainty inflation.

## B. Detailed Experimental Results

Due to space constraints in the main text, we present the comprehensive quantitative evaluations in this section.

Table 6 provides a granular breakdown of the scale-partitioned performance, serving as an extended version of Table 1. This detailed view further corroborates the effectiveness of our method in mitigating the long-tailed scale imbalance, particularly in the safety-critical *Far* and *Close* regimes.

Additionally, we report the complete benchmarking results under environmental corruptions in Table 7 (corresponding to Table 2 in the main text). These results cover a broader range of noise severities and corruption types, consistently demonstrating the superior stability of the proposed Decoupled Minimax Optimization framework against aleatoric uncertainty.

**CVaR-DRO Implementation Details.** We implement CVaR-DRO following a CVaR-style worst-case optimization at the image level. Specifically, for each training batch, we first compute the detection loss independently for each image,

including bounding box regression (CIoU), classification, and objectness terms. To avoid bias toward images containing more objects, the per-image loss is normalized by the number of ground-truth targets before difficulty ranking.

Given a batch of size $B$, we select the top-$k$ hardest samples according to the normalized loss, where $k = \max(1, \lfloor \alpha B \rfloor)$ and $\alpha = 0.2$. The final CVaR-DRO loss is computed as the mean of the original (unnormalized) losses over these selected samples, ensuring that optimization focuses on challenging cases without disproportionately amplifying multi-object images. This formulation corresponds to a CVaR objective that emphasizes the worst-performing fraction of the batch.

To stabilize training, we employ a warm-up strategy in which standard empirical risk minimization (ERM) is used for the first three epochs, after which CVaR-DRO is activated. All other training settings, including network architecture, optimizer, learning rate schedule, data augmentation, and training budget, are kept identical to the ERM baseline. We tune the CVaR-DRO hyperparameter $\alpha$ on the validation set and report the best-performing configuration.

We note that in UAV-on-UAV interception, many hard samples are dominated by strong aleatoric noise (e.g., motion blur or extreme small-scale targets), under which worst-case reweighting tends to overemphasize noise-dominated examples and can degrade overall stability.

*Table 6.* Distributional robustness across distance(scale) and viewpoint groups. We report mAP@0.50, mAP@0.75, and mAP@0.50:0.95 on stratified test subsets. Best results are **bolded**, and second-best results are underlined.

| Test Subset | Metric | Baseline | Focal | LS | DRO | PC | Ours |
|---|---|---|---|---|---|---|---|
| Test Set (Overall) | mAP@50 | 94.64 | 93.48 | 95.25 | 92.25 | 95.35 | **96.10** |
| | mAP@75 | 62.87 | 52.68 | 65.35 | 45.30 | 66.18 | **68.80** |
| | mAP@50:95 | 56.72 | 51.91 | 58.19 | 48.56 | 58.66 | **60.49** |
| *Scale Groups* | | | | | | | |
| Test Far (Small) | mAP@50 | 90.98 | 86.76 | 91.29 | 83.66 | 91.84 | **92.75** |
| | mAP@75 | 43.59 | 28.42 | 44.13 | 20.29 | **46.91** | 46.45 |
| | mAP@50:95 | 47.02 | 40.51 | 47.80 | 35.55 | **48.91** | 48.62 |
| Test Medium | mAP@50 | 97.96 | 98.70 | 98.50 | **98.97** | 98.67 | 98.80 |
| | mAP@75 | 81.62 | 68.94 | 81.91 | 64.89 | 81.64 | **85.57** |
| | mAP@50:95 | 66.39 | 59.57 | 66.77 | 58.50 | 67.13 | **69.88** |
| Test Close (Large) | mAP@50 | 95.45 | 97.43 | 96.83 | 97.06 | 96.25 | **98.01** |
| | mAP@75 | 79.59 | 74.66 | 83.92 | 75.10 | 77.96 | **88.99** |
| | mAP@50:95 | 66.02 | 62.43 | 68.27 | 62.16 | 64.19 | **73.56** |
| *Viewpoint Groups* | | | | | | | |
| Test Up | mAP@50 | 96.36 | 95.33 | 97.03 | 94.21 | 97.13 | **97.58** |
| | mAP@75 | 62.24 | 53.44 | 62.92 | 46.62 | 64.24 | **65.79** |
| | mAP@50:95 | 56.99 | 52.96 | 58.18 | 49.66 | 58.75 | **59.60** |
| Test Front | mAP@50 | 91.23 | 91.22 | 92.06 | 89.62 | 92.66 | **93.18** |
| | mAP@75 | 60.01 | 47.42 | 63.84 | 39.33 | 62.68 | **68.11** |
| | mAP@50:95 | 54.31 | 49.30 | 56.09 | 45.42 | 56.23 | **59.22** |
| Test Down | mAP@50 | 94.50 | 93.02 | 94.80 | 91.76 | 94.98 | **95.93** |
| | mAP@75 | 67.18 | 56.46 | 69.95 | 49.35 | 71.57 | **73.82** |
| | mAP@50:95 | 58.76 | 53.16 | 59.84 | 49.95 | 60.67 | **62.88** |

## C. Closed-loop Tracking and Real-world Interception Evaluation

To further evaluate whether improved localization translates into more stable closed-loop behavior, we report two complementary control-oriented evaluations. First, we evaluate angular tracking errors in a closed-loop simulation setting, where the controller acts on the image-plane deviation between the target center and the camera principal point. Second, we conduct real-world UAV-on-UAV interception trials to provide system-level evidence under physical deployment conditions.

**Simulation-based angular tracking evaluation.** Since center deviations directly determine the angular correction commands used by the downstream controller, we convert pixel-level center offsets into angular tracking errors using the pinhole camera model. Let $W$ denote the image width and FOV denote the horizontal field of view of the onboard camera. The focal length in pixel units is computed as

$$f = \frac{W/2}{\tan(\text{FOV}/2)}. \tag{20}$$

*Table 7.* Comparison of robustness under Gaussian noise and motion blur. We report **mAP$_{50}$**, **mAP$_{75}$**, **mAP$_{50:95}$** (%, higher is better) and **Center MAE** (pixels, lower is better). **Bold** and underlined indicate the best and second-best results.

| Noise Type | Metric | ERM | Focal | DRO | LS | PC | Ours |
|---|---|---|---|---|---|---|---|
| *Gaussian Noise* | | | | | | | |
| $\sigma = 0.05$ | mAP$_{50}$ ↑ | 92.60 | 91.06 | 79.79 | 92.33 | 92.60 | **93.42** |
| | mAP$_{75}$ ↑ | 55.79 | 47.18 | 14.24 | 57.40 | 58.49 | **60.69** |
| | mAP$_{50:95}$ ↑ | 53.34 | 48.92 | 29.79 | 53.86 | 54.45 | **56.10** |
| | Center MAE ↓ | 4.06 | 4.40 | 5.42 | 3.67 | 3.81 | **3.31** |
| $\sigma = 0.10$ | mAP$_{50}$ ↑ | 86.36 | 86.80 | 68.74 | 85.67 | 85.62 | **87.69** |
| | mAP$_{75}$ ↑ | 45.87 | 37.26 | 8.66 | 43.36 | 44.96 | **49.51** |
| | mAP$_{50:95}$ ↑ | 46.69 | 42.98 | 23.14 | 45.41 | 46.34 | **49.36** |
| | Center MAE ↓ | 4.34 | 4.83 | 6.05 | 4.27 | 4.33 | **3.77** |
| $\sigma = 0.15$ | mAP$_{50}$ ↑ | 77.35 | 77.79 | 55.30 | 76.61 | 75.94 | **79.53** |
| | mAP$_{75}$ ↑ | 35.53 | 27.20 | 4.26 | 32.45 | 30.63 | **38.53** |
| | mAP$_{50:95}$ ↑ | 39.60 | 35.76 | 16.87 | 38.00 | 37.25 | **41.49** |
| | Center MAE ↓ | 4.91 | 5.58 | 6.47 | 4.84 | 4.88 | **4.36** |
| $\sigma = 0.20$ | mAP$_{50}$ ↑ | 67.28 | 66.69 | 39.54 | 66.33 | 64.21 | **68.57** |
| | mAP$_{75}$ ↑ | 26.43 | 18.12 | 2.20 | 22.15 | 21.16 | **30.04** |
| | mAP$_{50:95}$ ↑ | 32.21 | 27.83 | 10.59 | 30.17 | 28.88 | **33.78** |
| | Center MAE ↓ | 5.37 | 6.28 | 6.81 | 5.41 | 5.42 | **4.79** |
| *Motion Blur* | | | | | | | |
| $k = 3$ | mAP$_{50}$ ↑ | 93.37 | 92.53 | 85.59 | 94.05 | 93.87 | **94.94** |
| | mAP$_{75}$ ↑ | 61.14 | 48.57 | 23.97 | 62.43 | 62.58 | **64.61** |
| | mAP$_{50:95}$ ↑ | 55.70 | 49.80 | 37.37 | 56.38 | 56.22 | **58.23** |
| | Center MAE ↓ | 3.80 | 4.26 | 5.09 | 3.61 | 3.82 | **3.33** |
| $k = 5$ | mAP$_{50}$ ↑ | 90.26 | 88.46 | 83.11 | 91.59 | 89.95 | **91.72** |
| | mAP$_{75}$ ↑ | 54.84 | 39.72 | 22.45 | 53.65 | 52.10 | **55.40** |
| | mAP$_{50:95}$ ↑ | 51.78 | 44.85 | 35.72 | 51.51 | 50.25 | **52.92** |
| | Center MAE ↓ | 3.96 | 4.56 | 5.31 | 3.90 | 4.09 | **3.60** |
| $k = 7$ | mAP$_{50}$ ↑ | 84.42 | 81.87 | 76.98 | **85.93** | 82.40 | 84.29 |
| | mAP$_{75}$ ↑ | **43.56** | 29.76 | 16.93 | 41.07 | 39.29 | 43.41 |
| | mAP$_{50:95}$ ↑ | 45.00 | 38.68 | 31.33 | 44.86 | 42.12 | **45.10** |
| | Center MAE ↓ | 4.30 | 5.08 | 5.79 | 4.25 | 4.64 | **3.96** |
| $k = 9$ | mAP$_{50}$ ↑ | 76.03 | 73.39 | 67.81 | **77.09** | 71.31 | 75.37 |
| | mAP$_{75}$ ↑ | 31.20 | 21.05 | 12.05 | 28.83 | 26.75 | **31.42** |
| | mAP$_{50:95}$ ↑ | **36.85** | 32.23 | 25.87 | 36.84 | 33.32 | 36.80 |
| | Center MAE ↓ | 4.59 | 5.58 | 6.28 | 4.77 | 5.27 | **4.35** |

Given the horizontal and vertical center offsets $(\Delta x, \Delta y)$ between the predicted target center and the image center, we define the corresponding angular deviations as

$$\theta_x = \arctan\left(\frac{\Delta x}{f}\right), \qquad \theta_y = \arctan\left(\frac{\Delta y}{f}\right). \tag{21}$$

The total angular error is then computed as

$$\theta_{\text{total}} = \sqrt{\theta_x^2 + \theta_y^2}. \tag{22}$$

All angular errors are reported in degrees. In addition, we define a frame as being in the *lock state* if the horizontal angular deviation satisfies $|\theta_x| \leq 5°$, and report the lock ratio as the percentage of locked frames over the full sequence.

As shown in Table 8, our method improves the lock ratio from $68.2\%$ to $84.8\%$, corresponding to an absolute gain of $16.6$ percentage points. The mean absolute horizontal angular error is reduced from $6.77°$ to $3.43°$, yielding a relative reduction of approximately $49.3\%$. The mean total angular error is also reduced from $8.48°$ to $5.46°$, corresponding to a relative reduction of approximately $35.6\%$. These results indicate that the proposed training objective not only improves static detection performance, but also improves target centering stability under closed-loop control in simulation.

**Real-world interception trials.** To complement the simulation-based angular tracking evaluation, we further conduct real-world UAV-on-UAV interception experiments in three outdoor environments. Each environment contains 20 interception attempts. An attempt is counted as successful if the pursuing UAV completes the interception procedure according to the predefined system-level success criterion.

*Table 8.* Simulation-based closed-loop angular tracking results. Lower angular errors indicate more stable target centering, while a higher lock ratio indicates more reliable closed-loop tracking.

| Method | Lock Ratio ↑ (%) | Mean Abs. $\theta_x$ ↓ (°) | Mean Signed $\theta_x$ (°) | Mean Abs. $\theta_y$ ↓ (°) | Mean Signed $\theta_y$ (°) | Mean Total Angular Error ↓ (°) |
|---|---|---|---|---|---|---|
| ERM | 68.2 | 6.77 | -2.24 | 3.41 | 1.03 | 8.48 |
| Ours | **84.8** | **3.43** | **0.32** | **3.40** | **-0.94** | **5.46** |

*Table 9.* Real-world UAV-on-UAV interception results across different outdoor environments.

| Environment | Attempts | Successes | Failures | Success Rate |
|---|---|---|---|---|
| Open large grassland | 20 | 19 | 1 | 0.95 |
| Hard-surface plaza | 20 | 18 | 2 | 0.90 |
| Small grassy area | 20 | 18 | 2 | 0.90 |
| Total / Average | 60 | 55 | 5 | 0.917 |

*Table 10.* Sensitivity analysis of scale partition boundaries. Hard partition and overlap grouping obtain similar performance, indicating that the method is not sensitive to exact boundary placement.

| Test Subset | Hard mAP@0.50 | Hard mAP@0.50:0.95 | Overlap mAP@0.50 | Overlap mAP@0.50:0.95 |
|---|---|---|---|---|
| Overall | 96.10 | 60.49 | 96.00 | 59.71 |
| Small / Far | 92.75 | 48.62 | 91.93 | 48.70 |
| Medium | 98.80 | 69.88 | 99.03 | 68.67 |
| Large / Close | 98.01 | 73.56 | 98.79 | 72.16 |

As shown in Table 9, the system achieves 55 successful interceptions out of 60 attempts, corresponding to an overall success rate of 91.7%. The success rates remain consistently high across open grassland, hard-surface plaza, and small grassy-area environments. These real-world trials provide complementary system-level evidence that stable image-center localization is practically relevant for UAV interception. In our closed-loop system, the aligning and reaching stages explicitly compute control commands from the target's image-center offsets; therefore, reducing center deviation directly contributes to more stable target alignment and improves the reliability of the interception process.

## D. Scale Partition Sensitivity

To evaluate whether the method is sensitive to hard partition boundaries, we compare the default hard partition with an overlap-grouping variant, where samples near the boundary are allowed to contribute to adjacent scale groups. The results are shown in Table 10.

As shown in Table 10, overlap grouping produces results close to the default hard partition across all test subsets. The overall mAP@0.50 changes from 96.10 to 96.00, and the overall mAP@0.50:0.95 changes from 60.49 to 59.71. Similar trends are observed for the small/far, medium, and large/close subsets. These results suggest that the proposed scale-partitioned training is robust to the exact placement of scale boundaries, and the main performance gains are not caused by a particular hard threshold choice.

## E. Comparison with Additional Recent Method

To comprehensively evaluate the performance and robustness of our proposed framework, we extend our comparative analysis to include MEDRO (Jeong et al., 2026), a recent state-of-the-art method for out-of-distribution generalization. We assess both methods under structured-shift conditions and against various image corruptions.

As detailed in Table 11, our method (SPUR) demonstrates strong competitiveness against MEDRO, achieving superior mAP@0.50 in the overall setting and across the majority of structured-shift subsets. Furthermore, under corruption evaluations involving Gaussian noise and motion blur, our approach exhibits significant robustness. Notably, it consistently outperforms MEDRO in Center MAE across all noise levels. These findings further substantiate the efficacy and reliability of our interception-specific design under challenging deployment conditions.

*Table 11.* Performance comparison between MEDRO and our method under structured-shift and corruption settings. The best results are highlighted in **bold**.

*(a) Structured-Shift Evaluation (mAP@0.50 ↑)*

| Method | Overall | Small / Far | Medium | Large / Close | Up | Front | Down |
|---|---|---|---|---|---|---|---|
| MEDRO | 95.94 | 92.49 | 98.72 | 97.46 | 97.48 | **94.07** | 95.27 |
| Ours (SPUR) | **96.10** | **92.75** | **98.80** | **98.01** | **97.58** | 93.18 | **95.93** |

*(b) Corruption Robustness Evaluation*

| Method | Gaussian Noise | | | | Motion Blur | | | |
|---|---|---|---|---|---|---|---|---|
| | Level 0.05 | Level 0.10 | Level 0.15 | Level 0.20 | Level 3 | Level 5 | Level 7 | Level 9 |
| mAP@0.50 ↑ | | | | | | | | |
| MEDRO | 93.06 | 86.87 | 78.51 | 66.80 | 94.35 | 90.87 | 83.27 | 72.53 |
| Ours (SPUR) | **93.42** | **87.69** | **79.53** | **68.57** | **94.94** | **91.72** | **84.29** | **75.37** |
| Center MAE ↓ | | | | | | | | |
| MEDRO | 3.95 | 4.46 | 5.06 | 5.82 | 3.58 | 3.87 | 4.25 | 4.80 |
| Ours (SPUR) | **3.31** | **3.77** | **4.36** | **4.79** | **3.33** | **3.60** | **3.96** | **4.35** |

