# OpenReview forum: "SPUR: Scale-Partitioned Uncertainty Rectification for Robust UAV-on-UAV Interception"
_ICML.cc/2026/Conference — ICML 2026 regular_

### Official Review · Reviewer_S6PZ · 2026-02-15

**Soundness:** 3
**Presentation:** 3
**Significance:** 3
**Originality:** 3
**Overall Recommendation:** 4
**Confidence:** 3

**Summary:**

This paper targets a very practical UAV-on-UAV interception setting where scale drift/imbalance and flight-induced visual noise break standard ERM training. It proposes a scale-partitioned minimax (GroupDRO-style) objective with a KL-divergence uncertainty set (Eq. 1), plus a decoupled aleatoric-uncertainty rectification to avoid “evasion” in coupled minimax+uncertainty, and a control-aligned center penalty for better downstream stability. Experiments on the UAV-MultiView benchmark (20,250 images) and corruption tests, together with embedded deployment / flight-test evidence, support the claimed robustness and real-time feasibility.

**Compliance With Llm Reviewing Policy:**

Affirmed.

**Key Questions For Authors:**

1. Eq. (1) KL-DRO vs. scale-balanced averaging: Why use the KL-divergence uncertainty set / inner maximization in Eq. (1) instead of simply averaging losses equally across the K scale subsets? Do you have an explicit baseline comparison? Impact: If KL-DRO is clearly motivated and empirically better than scale-balanced averaging, it strengthens the core contribution; otherwise it looks unnecessarily complex.

2. Related work / missing citation + scale tracking:
Sec. 2.3 contains “DRO (?)”. Please provide the intended citations and clarify how your formulation relates to standard KL-ball DRO / GroupDRO. Also, please add/position against prior scale-aware visual tracking work and explain the difference from your approach.
Impact: Clear positioning and complete citations increase confidence in rigor/novelty; lack of this weakens originality and presentation.

3. Impact / dual-use discussion:
Given the UAV interception context, can you expand the impact statement with concrete safeguards or release/usage constraints?
Impact: A credible mitigation discussion addresses a key concern; lack of it weakens the paper’s responsible framing.

**Limitations:**

No. The impact/limitations discussion is too brief for a UAV interception setting (it states that no societal consequences “must be specifically highlighted”).

**Strengths And Weaknesses:**

### Strength

- Soundness: The method is technically coherent and the paper explicitly validates a key failure mode (“evasion”) of coupled minimax+uncertainty and shows the decoupled design restores worst-group performance.

- Significance: Addresses a real deployment bottleneck in aerial interception: scale imbalance + motion blur/noise under tight compute/latency constraints, and evaluates on embedded hardware with a control-relevant metric (Center MAE).

- Empirical strength: Experiments are fairly comprehensive (group robustness, corruption robustness, ablations, and practical pipeline/flight-test evidence), not just standard perception benchmarks.

- Originality: The main novelty is a pragmatic combination—scale-partitioned minimax + uncertainty used as a gradient rectifier (decoupled) + control-aligned loss—well motivated by the UAV interception setting rather than generic detection.

### Weakness

- Motivation/positioning of the KL-divergence uncertainty set (Eq. 1) could be clearer. The paper states the KL-ball formulation “ensures each scale regime contributes equally”, but it does not clearly explain why this KL-based inner maximization is preferable to simply averaging losses across K scale subsets (your suggested baseline), or what behavior changes when moving from uniform group averaging to worst-case reweighting inside each group.

- Related work has a concrete gap / incomplete citation. In Sec. 2.3 it literally contains “Distributionally Robust Optimization (DRO) (?)”, suggesting a missing/placeholder reference. Also, the discussion of “scale problems” could better acknowledge classic visual tracking work on scale/aspect-ratio variation [1-3], since scale handling has been extensively studied in tracking. [1] integrating boundary and center correlation filters for visual tracking with aspect ratio variation. In ICCVW 2017. [2] Accurate scale estimation for robust visual tracking. BMCW 2014. [3] Automatic failure recovery and re-initialization for online uav tracking with joint scale and aspect ratio optimization. IROS 2020.

- Impact statement is likely under-discussed for an anti-UAV interception context. The current statement says there are societal consequences, “none … must be specifically highlighted,” which feels too brief given the obvious dual-use angle.

---

> ### Author Rebuttal · Authors · 2026-03-30
>
> We thank you for recognizing the important problem, the practical relevance of the setting, and the reasonable design of our scale-partitioned robust optimization framework. Your comments are highly helpful for improving the clarity, positioning, and responsible framing of the paper. We address the concerns below.
>
> ---
>
> #### **W1&Q1: Clarification of Eq. (1) and the role of the inner maximization**
> Thank you for raising this point. We clarify that our method is not a generic KL-DRO formulation. In Eq. (1), the KL term only defines the uncertainty set for the inner reweighting in our scale-partitioned minimax objective.
>
> By "scale-balanced averaging," we refer to averaging losses within each scale group with equal weights. The limitation is that it treats all samples in a group uniformly, so high-loss samples (e.g., due to clutter, blur, or viewpoint changes) are diluted by many easy ones. In contrast, our implementation induces an intra-group adversarial reweighting over per-sample losses, shifting the optimization toward harder samples within each scale regime.
>
> Therefore, the key difference from averaging is not cross-group balancing, but hardness-aware optimization within each group, which is crucial in our setting where samples within the same scale can still vary substantially in difficulty. We will revise the paper to clarify this distinction.
>
> #### **W2&Q2: Missing citation in Related Works and scale-aware tracking literature**
> We apologize for the mistake we made in Section 2.3. We will explicitly cite **GroupDRO** [1] in the revision. We clarify that our formulation is a **scale-partitioned KL-ball DRO objective** tailored to the dynamic scale imbalance of UAV interception, rather than a generic application of robust optimization.
>
> We will also expand the related-work discussion to include the classic visual-tracking literature on scale and aspect-ratio handling that you mentioned. These prior works are important, but their focus is different from ours: they mainly address online scale or aspect-ratio estimation for a tracked target over time, often in correlation-filter-style or sequential tracking settings, whereas our method addresses dataset-level scale imbalance and distribution shift in a real-time single-stage detection framework for interception. We will make this distinction clearer so that the novelty and positioning of our approach are more precise.
>
> ***References:***
>
> *[1] Sagawa, Shiori, et al. "Distributionally robust neural networks for group shifts: On the importance of regularization for worst-case generalization." arXiv preprint arXiv:1911.08731 (2019).*
>
> #### **W3&Q3: Impact statement and dual-use concerns**
>
> We agree that the current impact statement is too brief for a UAV interception paper. Our intended motivation is defensive and safety-oriented, such as protecting civilian airspace or critical infrastructure from unauthorized drones, but we recognize that the same perception pipeline could be misused in harmful settings. In the revision, we will substantially expand the Impact Statement to acknowledge this dual-use risk and describe concrete safeguards. In particular, we will clarify that any public release will focus on the **perception dataset and research code for training and evaluation**, and we will include explicit usage restrictions **against any non-research purpose usage**. We agree that responsible framing is essential here, and we will revise this section accordingly.
>
> ---
>
> We thank you again for the valuable feedback. These comments further support the need to clarify the theoretical positioning, related-work distinction, and responsible framing of our method. We will incorporate these improvements into the revised version.

---

> > ### Author Rebuttal · Reviewer_S6PZ · 2026-03-31
> >
> > The rebuttal addresses most of my main concerns and improves the clarity of the paper. One point remains only partially addressed: my question about whether the KL-based formulation performs better than a simpler scale-balanced averaging baseline was answered conceptually, but not with an explicit empirical comparison. Overall, the rebuttal is thoughtful and sufficient to maintain my current score, though not enough to change

---

> > > ### Author Response · Authors · 2026-04-04
> > >
> > > We thank the reviewer for the follow-up. To directly address your remaining question, we implemented a simpler scale-balanced averaging baseline that keeps the same $K$ scale groups as Eq. (1), but removes the inner maximization and uses equal weighting across groups without adversarial reweighting within each group.
> > >
> > > | Test Subset | Scale-Balanced Averaging | Ours |
> > > |---|:---:|:---:|
> > > | Overall | 94.68 | **96.10** |
> > > | Small / Far | 90.06 | **92.75** |
> > > | Medium | 97.75 | **98.80** |
> > > | Large / Close | 98.00 | **98.01** |
> > > | Up | 96.23 | **97.58** |
> > > | Front | 93.07 | **93.18** |
> > > | Down | 93.42 | **95.93** |
> > >
> > > These results show that our method consistently improves over the simpler scale-balanced averaging baseline on the overall set and on all structured-shift subsets. This suggests that the gain is not only from balancing the scale groups, but also from the within-group adversarial reweighting induced by the inner maximization. We will include this explicit empirical comparison in the revision.

---

### Official Review · Reviewer_dvbf · 2026-03-03

**Soundness:** 3
**Presentation:** 3
**Significance:** 4
**Originality:** 3
**Overall Recommendation:** 3
**Confidence:** 3

**Summary:**

This paper addresses the issues of scale drift, scale imbalance, and flight-induced noise in unmanned aerial vehicle (UAV) on UAV autonomous interception tasks. It proposes a scale-partitioned minimax robust optimization framework (SPUR). This method achieves robust modeling for different distance intervals and complex environmental disturbances by partitioning the training data according to the target scale and combining uncertainty correction with control-aware localization loss. Additionally, the authors introduce a decoupled minimax strategy to prevent the uncertainty mechanism from interfering with the mining of difficult samples. Experimental results show that this method significantly improves the stability of detection and localization under distribution drift and visual degradation conditions and can be implemented in real-time at high frame rates on embedded platforms.

**Compliance With Llm Reviewing Policy:**

Affirmed.

**Final Justification:**

The author provided a rather detailed response to my question. However, I still decided to maintain the original division.

**Key Questions For Authors:**

I still hope that the author can more clearly explain the direct correlation between the methods adopted and the research objectives, as well as how these equations can effectively express the author's ideas.

**Limitations:**

The author's motivation is very clear. However, the paper is not very comprehensive and seems a bit rushed, with many aspects not being well explained.

**Strengths And Weaknesses:**

Strengths:

1 The paper not only points out the conflict between scale imbalance and accidental uncertainty but also reveals from the perspective of the optimization mechanism that the "coupling" of minimax and uncertainty modeling will lead to an evasion phenomenon, and proposes a decoupling strategy to solve it. This complete logical chain from theoretical conflict to mechanism design makes the method have strong interpretability and theoretical depth, rather than just being an engineering improvement.

2 The method only affects the training objective function without altering the inference structure, thus hardly increasing the inference cost. It achieves real-time performance of over 100 FPS on the embedded NVIDIA Orin NX, and significantly enhances robustness under distribution drift and visual disturbances, demonstrating a high degree of alignment between the algorithm design and the actual deployment scenarios of unmanned aerial vehicles. This balance of "high robustness + high real-time" has practical application value in air interception tasks.


Weaknesss:

1 The variables in Eq. 1 are not fully explained. Moreover, the intended purpose of this formula and how it achieves that purpose are not clearly connected to scale optimization. Additionally, the criteria for scale classification are not specified. For UAVs of different sizes, does this size classification need adjustment? Could the grouping method mentioned in the paper have any impact near the classification boundaries?

2 Regarding the use of uncertainty correction to address flight-induced noise, what is the underlying principle here? Noise in some images reduces tracking accuracy, and the author naturally mitigates the impact of inherent noisy samples through exponential decay, which only reduces the influence of certain images. However, if there is a prolonged sequence of noisy images, this method may actually lead to losing track of the target. In other words, the author seems to be focusing on ignoring highly noisy images rather than addressing the issue of tracking inaccuracies caused by noise.

3 The comparison methods are not up-to-date; methods from the past three years should be added for comparison.

---

> ### Author Rebuttal · Authors · 2026-03-30
>
> We thank you for recognizing the important problem, the interpretability of our method, and the practical value of robust real-time UAV interception. We address the concerns below.
>
> ---
>
> #### **W1&Q1: Clarification of Eq. 1, its connection to scale optimization, and the effect of grouping boundaries**
>
> We agree that Eq. 1 and its connection to the scale objective should be clarified. In Eq. 1, $K$ is the number of scale groups, $S_k$ the samples in group $k$, $A_i$ the bounding-box area as scale proxy, $p_{k,0}$ the empirical distribution, and $\mathcal{P}_k$ the uncertainty set. The key idea is that scale partitioning limits mid-scale dominance, while the inner maximization emphasizes harder samples within each group. In practice, group contributions are aggregated with normalization over positive samples rather than explicit equal group weighting. We will revise Section 3.1 to make this connection between the physical scale-drift problem and Eq. 1 more explicit.
>
> Regarding the grouping criterion, our grouping is relative rather than absolute: the scale partitions are determined from normalized bounding-box areas in the training distribution, with uniform partitioning into $𝐾$ groups over $[0,1]$ in our implementation. Therefore, if the physical UAV size changes in another deployment scenario, the grouping thresholds shift with the new data distribution rather than being fixed to one UAV type.
>
> We also agree that boundary effects deserve clarification. To examine this, we conducted an additional overlap-grouping experiment. The results shown below suggest that the impact of the boundary grouping method near the classification boundaries is almost negligible.
>
> | Test Subset | Hard Partition mAP@0.50 | Hard Partition mAP@0.50:0.95 | Overlap Grouping mAP@0.50 | Overlap Grouping mAP@0.50:0.95 |
> | --- | --- | --- | --- | --- |
> | Overall | 96.10 | 60.49 | 96.00 | 59.71 |
> | Small / Far | 92.75 | 48.62 | 91.93 | 48.70 |
> | Medium | 98.80 | 69.88 | 99.03 | 68.67 |
> | Large / Close | 98.01 | 73.56 | 98.79 | 72.16 |
>
> #### **W2: Principle of uncertainty rectification under flight-induced noise**
>
> Our uncertainty rectification is intended to improve training dynamics, not to discard noisy images. In UAV interception, images during pursuit can remain highly noisy and hard, and thus cannot be removed during training. Such samples may yield large losses due to irreducible aleatoric noise rather than informative structure; if aggressively upweighted by minimax optimization, they can destabilize learning. Our uncertainty term reduces their regression gradients, while the decoupled minimax strategy still emphasizes structurally difficult but reliable samples.
>
> During inference, the model predicts on every frame; we do not ignore noisy images online. We will revise Section 3.2 and 3.4 to clarify this distinction.
>
> #### **W3: Comparison with more recent methods**
>
> We agree that the comparison set should be strengthened. Following your suggestion, we additionally evaluated a recent state-of-the-art method, MEDRO [1], and provided the results of the structured-shift evaluation and corruption robustness evaluation below.
>
> | Test Subset | MEDRO mAP@0.50 | Ours mAP@0.50 |
> |---|---|---|
> | Overall | 95.94 | **96.10** |
> | Small / Far | 92.49 | **92.75** |
> | Medium | 98.72 | **98.80** |
> | Large / Close | 97.46 | **98.01** |
> | Up | 97.48 | **97.58** |
> | Front | **94.07** | 93.18 |
> | Down | 95.27 | **95.93** |
>
> | Noise Type | Level | MEDRO mAP@0.50 | MEDRO Center MAE | Ours mAP@0.50 | Ours Center MAE |
> |---|---|---|---|---|---|
> | Gaussian | 0.05 | 93.06 | 3.9515 | **93.42** | **3.31** |
> | Gaussian | 0.10 | 86.87 | 4.4602 | **87.69** | **3.77** |
> | Gaussian | 0.15 | 78.51 | 5.0584 | **79.53** | **4.36** |
> | Gaussian | 0.20 | 66.80 | 5.8229 | **68.57** | **4.79** |
> | Motion Blur | 3 | 94.35 | 3.5825 | **94.94** | **3.33** |
> | Motion Blur | 5 | 90.87 | 3.8732 | **91.72** | **3.60** |
> | Motion Blur | 7 | 83.27 | 4.2525 | **84.29** | **3.96** |
> | Motion Blur | 9 | 72.53 | 4.7963 | **75.37** | **4.35** |
>
> These results show that SPUR remains competitive with MEDRO and is stronger in the **overall setting**, most **structured-shift subsets**, and under both **Gaussian noise** and **motion blur**, especially in **Center MAE**. This further supports the effectiveness of our interception-specific design. We will add more recent baselines and more complete comparisons in the revision.
>
> ***References:***
>
> *[1] Jeong, Jinyong, et al. "Multi-Expert Distributionally Robust Optimization for Out-of-Distribution Generalization." The Thirty-ninth Annual Conference on Neural Information Processing Systems, 2025*
>
> ---
>
> We thank you for the valuable feedback. These comments further support the need to clarify the scale-partition mechanism, the role of uncertainty rectification, and the empirical positioning of SPUR. We will incorporate these clarifications and additional results into the revised version.

---

> > ### Author Rebuttal · Reviewer_dvbf · 2026-04-04
> >
> > Regarding the second question, the authors did not provide a clear answer. You merely reduced the weights rather than effectively solving the problem—this is not a solution. Noisy images are inherent and persist over long periods.

---

> > > ### Author Response · Authors · 2026-04-07
> > >
> > > Thank you for the follow-up. You are correct that **uncertainty rectification is not a direct solution for reconstructing missing information from persistently corrupted frames**. That is not the goal of our method.
> > >
> > > In UAV interception, noisy observations such as motion blur, occlusion, and rapid viewpoint change are **not rare outliers or corrupted annotations**, but a **common and unavoidable part of the task itself**. Therefore, neither treating them as equally reliable and aggressively upweighting them, nor discarding them entirely, is appropriate. The former can destabilize minimax optimization, while the latter creates a mismatch between training and real deployment.
> > >
> > > Our point is more specific. In the optimization loss of Eq. (6), the uncertainty attenuation is applied only to the **regression branch** inside $L_{\text{Uncertainty}}$. The **classification loss** $L_{\text{cls}}$ and the **control-aware term** $L_{\text{Control}}$ are **not attenuated**. Therefore, noisy frames are **not ignored**: they still contribute fully to target recognition and center-alignment learning, while only the unreliable boundary-regression gradients are reduced.
> > >
> > > This distinction is important under persistent motion blur or sensor noise. In such cases, forcing the model to fit a precise bounding box around an intrinsically ambiguous target boundary can corrupt the regression updates. By contrast, SPUR keeps learning from these frames through the **unattenuated classification and control signals**, while preventing the noisy regression term from dominating optimization. In this sense, the method does not "solve" persistent noise by removing the corruption itself; rather, it improves robustness by **separating target recognition and control-relevant centering from unreliable boundary fitting**.
> > >
> > > This mechanism is especially important under minimax reweighting. Without uncertainty rectification, the inner maximization can repeatedly over-focus on persistent noisy frames and mistake them for informative hard examples. With SPUR, noisy frames remain in both training and inference, but their **regression gradients are modulated rather than allowed to dominate the update**.
> > >
> > > This interpretation is also consistent with our corruption experiments in **Table 2 of our paper**, where our method shows **consistently better robustness and lower center localization error** under increasing Gaussian noise and motion blur, rather than collapsing under sustained corruption. We will clarify this point more explicitly in the revision.

---

### Official Review · Reviewer_sPqW · 2026-03-10

**Soundness:** 3
**Presentation:** 2
**Significance:** 2
**Originality:** 3
**Overall Recommendation:** 3
**Confidence:** 3

**Summary:**

This paper presents SPUR, a scale-aware robust optimization framework for UAV-on-UAV interception under dynamic scale variations and visual noise. The proposed method performs scale-partitioned minimax optimization to improve robustness across different target scales, and incorporates uncertainty-rectified regression and a control-aware localization objective. The paper also introduces the UAV-MultiView dataset for aerial interception scenarios. Experiments demonstrate competitive performance compared with several baselines.

**Compliance With Llm Reviewing Policy:**

Affirmed.

**Final Justification:**

The rebuttal addresses several of my main concerns and improves the clarity of the paper, particularly through the additional comparisons with simpler alternatives and the stronger empirical justification of the scale-partitioned design. While I still have some reservations about the overall evaluation breadth and impact, I am willing to raise my score to **weak reject**.

**Key Questions For Authors:**

Please refer to weaknesses above.

**Limitations:**

yes

**Strengths And Weaknesses:**

**Strengths:**
1. The paper addresses the problem of robust UAV detection for aerial interception, which is relevant for autonomous UAV systems.

2. The proposed framework combines scale-partitioned minimax optimization and uncertainty-aware regression, which appears to be a reasonable approach for handling scale imbalance and observation noise.

3. The experimental results appear promising and show improvements over several baseline methods on the UAV-MultiView dataset.

**Major weakness:**
1. Please further clarify the theoretical basis of the proposed Decoupled Minimax Optimization strategy, particularly why separating the proxy loss used for adversarial weighting from the optimization loss is necessary to prevent the uncertainty module from suppressing hard examples.

2. The UAV-MultiView dataset is relatively small compared with widely used benchmarks[1-3], which may limit the diversity of training samples. The paper also lacks detailed analysis on how well the collected data reflects real UAV interception dynamics. A more detailed comparison with existing UAV detection datasets would help better contextualize the proposed dataset.

[1] Anti-uav: A large-scale benchmark for vision-based uav tracking (IEEE TMM 2021)

[2] Anti-UAV410: A thermal infrared benchmark and customized scheme for tracking drones in the wild (IEEE TPAMI 2023)

[3] Detection and tracking meet drones challenge (IEEE TPAMI 2021)

3. The method partitions samples into discrete scale groups based on the object area $A\_i$. Since target scale varies continuously in UAV interception scenarios, such hard partitioning may introduce boundary effects for samples near the partition thresholds. It would be helpful to analyze how sensitive the method is to these partition boundaries, and whether soft or overlapping scale grouping could provide a more stable alternative.

4. The proposed loss formulation introduces multiple weighting hyperparameters (e.g., $\lambda\_1$ and $\lambda\_2$), but the paper lacks analysis of their influence on training and performance. This makes it unclear how sensitive the method is to hyperparameter choices.

**Minor weakness:**
1. Several prior works have explored related directions such as distributionally robust optimization, uncertainty-aware regression, and scale-aware detection. Since the proposed method also combines these components, it would be helpful to more clearly describe the differences from existing approaches and highlight the specific novelty of the proposed framework.

2. A citation appears to be missing in line 130.

---

> ### Author Rebuttal · Authors · 2026-03-30
>
> We thank you for recognizing the important problem, the practical relevance of robust UAV-on-UAV interception, and the reasonable design of our scale-partitioned robust optimization framework. We address the concerns below.
>
> ---
> #### **W1: Theoretical basis of the Decoupled Minimax Optimization**
> Thank you for highlighting this point. The theoretical analysis of this issue has been provided in Appendix A. In the coupled setting, if adversarial weights are computed from the uncertainty-rectified loss, the optimal uncertainty satisfies $s_i^* = \log \mathcal{L}_{\text{raw}}$, which weakens the hard-example signal and allows genuinely hard samples to evade adversarial weighting. Our decoupled strategy avoids this by computing adversarial weights from the **raw proxy loss** and using uncertainty only in the **optimization loss**. This preserves hard-example mining while still attenuating gradients from irreducible aleatoric noise. The same effect is also observed empirically in Table 4, where the coupled variant drops to **86.76%** on *Far*, while the decoupled version restores it to **92.18%**.
>
> #### **W2: Dataset scale, realism, and positioning of UAV-MultiView**
> We agree that UAV-MultiView should be positioned more clearly. Our goal is not to provide the largest UAV benchmark, but a task-specific interception benchmark where both drones are in motion, target scale drifts continuously during pursuit, and viewpoint changes are explicit. UAV-MultiView contains 20,250 annotated images and is structured along Far/Medium/Close and Up/Front/Down axes, which directly supports the stratified robustness analysis in our paper. We will **cite the three works** you mentioned above and add a clearer positioning discussion in the revision. A concise comparison is shown below.
>
> | Aspect | Anti-UAV [1] | Anti-UAV410 [2] | DTMD [3] | Ours |
> |---|---|---|---|---|
> | Main target | General UAV | Thermal UAV | Drone detection/tracking | **Interception** |
> | Geometry | Mostly static | Mostly static | Mixed | **Both UAVs moving** |
> | Scale focus | Not explicit | Not explicit | Partial | **Far / Medium / Close** |
> | View focus | General | General | General | **Up / Front / Down** |
> | Control focus | – | – | – | **✓** |
>
> #### **W3: Hard partitioning, boundary effects, and overlap-grouping**
> In the current paper, scale groups are formed from the target bounding-box area distribution using spectral clustering with $K=3$. To directly examine the effect of boundary handling, we conducted an additional overlap-grouping experiment. The results are shown below.
>
> | Test Subset | Hard Partition mAP@0.50 | Hard Partition mAP@0.50:0.95 | Overlap Grouping mAP@0.50 | Overlap Grouping mAP@0.50:0.95 |
> | --- | --- | --- | --- | --- |
> | Overall | 96.10 | 60.49 | 96.00 | 59.71 |
> | Small / Far | 92.75 | 48.62 | 91.93 | 48.70 |
> | Medium | 98.80 | 69.88 | 99.03 | 68.67 |
> | Large / Close | 98.01 | 73.56 | 98.79 | 72.16 |
>
> These results show that boundary effects are almost negligible and the method is not sensitive to partition boundaries.
>
> #### **W4: Sensitivity to weighting hyperparameters**
> We provide the sensitivity analysis of $(\lambda_2,\lambda_1)$ below.
>
> | λ2,λ1 | mAP@0.50 | Far | Medium | Close | G0.05-mAP | G0.05-MAE | MB5-mAP | MB5-MAE |
> |---|---|---|---|---|---|---|---|---|
> | 0.01,1.0 | 0.9575 | 0.9153 | 0.9867 | 0.9796 | 0.9288 | 3.6671 | 0.9092 | 3.9015 |
> | 0.001,1.0 | 0.9562 | 0.9125 | 0.9870 | 0.9787 | 0.9300 | 3.7100 | 0.9088 | 3.8987 |
> | 0.005,0.5 | 0.9580 | 0.9156 | 0.9883 | 0.9783 | 0.9234 | 3.7539 | 0.9106 | 3.9627 |
> | 0.005,2.0 | 0.9562 | 0.9112 | 0.9864 | 0.9836 | 0.9237 | 3.6902 | 0.9070 | 3.8912 |
> | 0.005,1.0 | 0.9610 | 0.9275 | 0.9880 | 0.9801 | 0.9342 | 3.3061 | 0.9172 | 3.6001 |
>
> These results show that the method is reasonably stable across a nontrivial range of $(\lambda_2$, $\lambda_1)$, and that our default setting (0.005,1.0) provides the best or near-best trade-off across overall accuracy, far-target robustness, and corruption robustness.
>
> #### **W5: Novelty relative to existing methods**
> We agree that the novelty should be stated more clearly. SPUR is **not** a simple combination of existing DRO, uncertainty-aware, and scale-aware methods. Its key novelty is identifying and resolving an **optimization conflict** in UAV interception: minimax reweighting amplifies noisy outliers, while directly coupling uncertainty into adversarial weighting lets hard examples evade reweighting. Our **Decoupled Minimax Optimization** resolves this issue. In addition, the method is **scale-partitioned** for pursuit-induced far/medium/close imbalance and includes a **control-aware localization term** aligned with downstream flight sensitivity. We will clarify this more explicitly in the revision.
>
> #### **W6: Missing citation**
> Thank you for pointing this out. We will add the missing citations in the revision.
>
> ---
> We thank you for the valuable feedback. We will incorporate these clarifications and additional results into the revised version.

---

> > ### Author Rebuttal · Reviewer_sPqW · 2026-04-02
> >
> > The rebuttal addresses part of my main concerns and improves the clarity of the paper. In particular, the authors provide additional analysis on the scale partition strategy, hyperparameter sensitivity, and dataset positioning, which help strengthen the empirical justification of the method.
> >
> > However, some points remain only partially addressed. For example, while the authors provide additional experiments on overlap grouping, it is still unclear whether the proposed scale-partition design is necessary compared to simpler alternatives (e.g., continuous scale-aware weighting). In addition, the proposed method relies on a scale-partitioned minimax objective, but it is unclear how much of the performance gain comes from the minimax formulation itself, compared to simpler alternatives such as uniform weighting across scale groups.
> >
> > Overall, the rebuttal addresses several concerns and improves clarity to some extent.

---

> > > ### Author Response · Authors · 2026-04-04
> > >
> > > Thank you for the follow-up. This is an important question. To directly address it, we implemented two simpler alternatives to our full scale-partitioned minimax design:
> > >
> > > 1. **Scale-Avg**: equal weighting across the $K$ scale groups, without inner maximization.
> > > 2. **Continuous**: continuous scale-aware weighting, without discrete scale-group minimax.
> > >
> > > We report the experiment results below.
> > >
> > > | Test Subset | Scale-Avg | Continuous | Ours |
> > > | --- | :---: | :---: | :---: |
> > > | Overall | 94.68 | 95.45 | **96.10** |
> > > | Small / Far | 90.06 | 91.25 | **92.75** |
> > > | Medium | 97.75 | 98.48 | **98.80** |
> > > | Large / Close | 98.00 | 97.95 | **98.01** |
> > > | Up | 96.23 | 96.64 | **97.58** |
> > > | Front | 93.07 | 93.11 | **93.18** |
> > > | Down | 93.42 | 94.77 | **95.93** |
> > >
> > > These results suggest two points. First, compared with **Scale-Avg**, the gain does not come only from balancing contributions across scale groups, but also from the minimax reweighting within each group. Second, compared with the **Continuous** alternative, the consistent improvement supports the value of the explicit scale-partitioned minimax design rather than a simpler continuous scale-aware weighting scheme. We will include these comparisons in the revision to make the contribution of each design choice clearer.

---

### Official Review · Reviewer_3ZAq · 2026-03-13

**Soundness:** 3
**Presentation:** 3
**Significance:** 4
**Originality:** 3
**Overall Recommendation:** 4
**Confidence:** 5

**Summary:**

The paper proposes a scale-partitioned uncertainty rectification framework to address the challenge of systematic scale drift in autonomous UAV-on-UAV interception, and introduces a new dataset featuring rich geometric and viewpoint variations

**Compliance With Llm Reviewing Policy:**

Affirmed.

**Final Justification:**

Thank authors for the comprehensive rebuttal. I will keep my score, but with higher confidence.

**Key Questions For Authors:**

- Could you provide direct closed-loop flight metrics to empirically validate the "control-aware" claim?
- Could you provide the rationale and a sensitivity analysis for the hard area thresholds, and clarify how the framework maintains generalization across different camera hardware?
- Please do a final proofreading pass to ensure all LaTeX citation keys are properly compiled (e.g., the missing citation around line 130), and all spellings are correct (e.g., "PHYSICAL PERSUIT" and "SCALE PATITIONING" in Figure 1)

**Limitations:**

The authors have not adequately discussed the limitations or potential negative societal impacts of their work. The current Impact Statement is too generic, given the application domain. Since the paper studies autonomous UAV-on-UAV interception, the work raises nontrivial concerns related to physical safety, misuse in security or military contexts, and regulatory compliance. In addition, the limitations section should more explicitly discuss the method's dependence on predefined scale partitions and its uncertain robustness under broader distribution shifts

**Strengths And Weaknesses:**

Pros

- The paper provides clear and logically consistent mathematical derivations when addressing the underlying conflict between minimax optimization and uncertainty estimation.
- The narrative is logically coherent. Additionally, the proposed framework diagram intuitively and comprehensively illustrates the algorithm architecture and the closed-loop system engineering.
- The research closely aligns with the real-world demands of the low-altitude economy and Anti-UAV scenarios. Coupled with the contribution of an open-source dataset, this work offers strong engineering guidance and significant potential for practical deployment.

Cons

- The paper employs preset, hard area thresholds to discretize targets into scale groups, but it does not sufficiently justify the criteria for selecting these thresholds. It remains unclear whether the existing conclusions would still hold if the partition boundaries were changed, or if the model were migrated to camera hardware with different focal lengths and resolutions.
- The paper highlights "control-aware" as a core contribution. However, the experimental validation currently relies on indirect, static visual metrics such as Center MAE. It lacks direct physical flight metrics

---

> ### Author Rebuttal · Authors · 2026-03-30
>
> We thank you for recognizing the important problem, the practical relevance of robust UAV-on-UAV interception, and the reasonable design of our scale-partitioned robust optimization framework. We address the concerns below.
>
> ---
>
> #### **W1&Q2: Scale partition thresholds, sensitivity, and generalization across hardware**
> Thank you for this important comment. We agree that the rationale behind the scale partitions should be stated more clearly. In our method, the scale grouping is not chosen manually by fixed semantic rules. As described in the paper, we partition samples using the target bounding-box area $A_i$ as a scale proxy, and in the current implementation, we use spectral clustering with $K=3$ to form the scale groups. We will revise the text to make this explicit and to better explain the connection between the scale-drift problem and Eq. (1).
>
> Regarding **generalization across different camera hardware**, our framework does not assume that one fixed set of partition boundaries is universal. The partitions are derived from the target-scale distribution of the training data in the deployment setting. Therefore, if focal length, resolution, or flight geometry changes, the grouping can be recomputed from the new scale distribution rather than transferred as fixed thresholds.
>
> We also provide an overlap-grouping experiment to analyze the sensitivity of the hard area threshold. The results are shown below.
>
> | Test Subset | Hard Partition mAP@0.50 | Hard Partition mAP@0.50:0.95 | Overlap Grouping mAP@0.50 | Overlap Grouping mAP@0.50:0.95 |
> | --- | :---: | :---: | :---: | :---: |
> | Overall | 96.10 | 60.49 | 96.00 | 59.71 |
> | Small / Far | 92.75 | 48.62 | 91.93 | 48.70 |
> | Medium | 98.80 | 69.88 | 99.03 | 68.67 |
> | Large / Close | 98.01 | 73.56 | 98.79 | 72.16 |
>
> These results show that boundary handling's effects are almost negligible, and the method is not sensitive to the hard boundary threshold. This does not change the main conclusion that scale-partitioned training improves robustness across distance regimes.
>
> #### **W2&Q1: Direct validation of the "control-aware" claim**
>
> We agree that direct closed-loop metrics are the strongest validation of the control-aware claim. In the current submission, we use Center MAE as a perception-level proxy because center deviations directly affect angular tracking error in UAV interception. In addition, we conducted real-world UAV-on-UAV interception experiments to serve as additional system-level evidence. The results are provided below.
>
> | Environment | Attempt | Success | Failure | Succ. Rate |
> | :--- | :---: | :---: | :---: | :---: |
> | Open large grassland | 20 | 19 | 1 | 0.95 |
> | Hard-surface plaza | 20 | 18 | 2 | 0.90 |
> | Small grassy area | 20 | 18 | 2 | 0.90 |
> | **Total & Average** | **60** | **55** | **5** | **0.917** |
>
> Importantly, in this closed-loop system, the aligning and reaching stages explicitly compute control commands from the target’s image-center offsets, which directly supports the practical relevance of center-stable localization for interception success. We will add these results as complementary closed-loop evidence for control-aware motivation in the revision.
>
> #### **W3: Proofreading, missing citation, and figure typos**
> Thank you for catching these issues. We will correct them in the revision.
>
> #### **W4: Broader impact and limitations**
> We agree that the current impact statement is too generic. Our intended motivation is defensive and safety-oriented, such as protecting civilian airspace or critical infrastructure from unauthorized drones, but we recognize that the same perception pipeline could be misused in harmful settings. In the revision, we will clarify that any public release will focus on the **perception dataset and research code for training and evaluation**, and we will include explicit usage restrictions against any **non-research purpose usage**.
>
> We will also expand the limitations discussion to state more clearly that the current framework depends on predefined scale partitions and that robustness under broader out-of-distribution shifts remains an open question.
>
> ---
>
> We thank you for the valuable feedback. These comments further support the need to clarify the rationale of scale partitioning, strengthen the control-aware validation, and improve the broader-impact discussion. We will incorporate these clarifications and additional results into the revised version.

---

> > ### Author Rebuttal · Reviewer_3ZAq · 2026-04-02
> >
> > Thanks to the authors. The rebuttal addresses most of my concerns. For W2&Q1, could the authors additionally report finer-grained control-related metrics (e.g., lock duration, angular error, or recovery-related statistics), and, if possible, compare them against relevant baselines in the same closed-loop setting?

---

> > > ### Author Response · Authors · 2026-04-06
> > >
> > > Thank you for the follow-up. To address this point, we additionally report finer-grained **closed-loop control metrics** under the **same closed-loop evaluation setting** for both our method and the ERM baseline.
> > >
> > > We convert image-plane center offsets into angular tracking errors using the pinhole camera model. Let $W$ denote the image width, the focal length in pixels is
> > >
> > > $$f=\frac{W/2}{\tan (\text{FOV}/2)}.$$
> > >
> > > Given horizontal and vertical center offsets $\Delta x,\Delta y$, we define
> > >
> > > $$\theta_x = \arctan\left(\frac{\Delta x}{f}\right), \quad \theta_y = \arctan\left(\frac{\Delta y}{f}\right),$$
> > >
> > > and the **Total Angular Error** as $\sqrt{\theta_x^2 + \theta_y^2}$.
> > >
> > > To reflect closed-loop tracking quality, we further define a frame to be in **lock state** if $|\theta_x|\leq 5^{\circ}$, and report the **Lock Ratio**, i.e., the percentage of locked frames over the full sequence.
> > >
> > > | Method | Lock Ratio ↑ | Mean Abs. $\theta_x$ (°) ↓ | Mean Signed $\theta_x$ (°) | Mean Abs. $\theta_y$ (°) ↓ | Mean Signed $\theta_y$ (°) | Mean Total Angular Error (°) ↓ |
> > > |---|---|---|---|---|---|---|
> > > | ERM | 68.2% | 6.77 | -2.24 | 3.41 | 1.03 | 8.48 |
> > > | **Ours** | **84.8%** | **3.43** | **0.32** | **3.40** | **-0.94** | **5.46** |
> > >
> > > These results show that the improvement is not only in static detection accuracy, but also in **closed-loop control quality**. Our method increases the lock ratio by **16.6 points**, reduces horizontal angular error by nearly **50%**, and lowers the total angular error from **8.48°** to **5.46°**. This provides direct evidence that the proposed control-aware design leads to more stable target centering in closed-loop execution. We will add these control-related results to the revision and extend the comparison to additional baselines in the same closed-loop setting.

---

### Decision · Program_Chairs · 2026-04-30

**Decision:**

Accept (regular)

**Comment:**

In this paper, the authors propose SPUR, a scale-partitioned robust optimization framework for UAV-on-UAV detection and interception. The framework comprises (1) a scale-partitioned KL-divergence minimax objective that partitions samples into K=3 scale groups (Far/Medium/Close) via spectral clustering on bounding-box area and applies group-wise adversarial reweighting; (2) a decoupled aleatoric uncertainty rectification, where adversarial weights are computed from a raw proxy loss while uncertainty modulates only the regression branch of the optimization loss, preventing the "evasion" where coupling would let hard examples escape reweighting; and (3) a control-aware localization loss penalizing image-center deviation. The authors also introduce UAV-MultiView, a 20,250-image dataset structured along Far/Medium/Close x Up/Front/Down axes. Evaluation covers structured shifts, corruption robustness (Gaussian noise, motion blur), embedded deployment on NVIDIA Orin NX (>100 FPS), and real-world UAV-on-UAV flight tests.

Post-rebuttal reviews split into two weak accepts (R-3ZAq with confidence 5, R-S6PZ) and two weak rejects (R-sPqW, R-dvbf, both confidence 3). Concerns covered scale-partition threshold sensitivity and hardware generalization; direct closed-loop validation of the "control-aware" claim (R-3ZAq); hyperparameter sensitivity and theoretical justification of the decoupled formulation (R-sPqW); the principle of uncertainty rectification under persistent flight-induced noise (R-dvbf); missing recent SOTA baselines (R-dvbf); KL-DRO motivation versus simpler scale-balanced averaging (R-S6PZ); UAV-MultiView size relative to standard benchmarks (R-sPqW); and thin dual-use impact discussion (R-3ZAq, R-S6PZ).

Throgh the rebuttal, authors presented an overlap-grouping experiment showing partition-boundary effects are negligible; closed-loop control metrics with substantial improvements (Lock Ratio 84.8% vs 68.2% ERM, total angular error 5.46° vs 8.48°); real-world flight tests across three environments (60 trials, 91.7% success); a Scale-averaging vs Continuous vs SPUR ablation (94.68 / 95.45 / 96.10 mAP@0.50) confirming the inner maximization contributes beyond simple scale balancing; a hyperparameter sensitivity sweep; a MEDRO (NeurIPS 2025) comparison showing stronger Center MAE under corruption; clarification that uncertainty attenuation applies only to regression gradients while classification and control-aware terms remain unattenuated; and an expanded impact statement with usage restrictions. R-3ZAq maintained weak accept; R-S6PZ was satisfied after the Scale-Avg experiment; R-sPqW maintained weak reject on remaining evaluation-breadth reservations; R-dvbf maintained weak reject, finding the noise-handling explanation insufficient.

The contribution of a control-aligned, decoupled minimax + uncertainty framework with real-world flight-test validation is useful for the target deployment scenario, and the rebuttal additions are substantive. The most confident reviewer recommends acceptance, and the new closed-loop metrics directly substantiate the control-aware claim that was originally validated only through static proxies. R-dvbf's remaining objection reads as disagreement with the design intent which I feel was addressed by the authors. The authors' uncertainty mechanism only attenuates bounding-box regression gradients, while still letting classification and center-alignment learn from noisy frames, which is a reasonable way to handle persistent noise that cannot be denoised or discarded, and the paper's corruption experiments back it up empirically. R-sPqW's evaluation-breadth concern is also noted, though the rebuttal additions go some way toward it. I recommend acceptance, and encourage the authors to integrate the closed-loop control metrics, real-world flight-test results, Scale-Avg/Continuous baselines, MEDRO comparison, and expanded impact discussion into the main text.